# Invariant Feature Coding using Tensor Product Representation

**Yusuke Mukuta**     *mukuta@mi.t.u-tokyo.ac.jp*
*The University of Tokyo, RIKEN*

**Tatsuya Harada**     *harada@mi.t.u-tokyo.ac.jp*
*The University of Tokyo, RIKEN*

**Reviewed on OpenReview:** *https://openreview.net/forum?id=uv32JOdQuh*

## Abstract

In this study, a novel feature coding method that exploits invariance for transformations represented by a finite group of orthogonal matrices is proposed. We prove that the group-invariant feature vector contains sufficient discriminative information when learning a linear classifier using convex loss minimization. Based on this result, a novel feature model that explicitly considers group action is proposed for principal component analysis and k-means clustering, which are commonly used in most feature coding methods, and global feature functions. Although the global feature functions are in general complex nonlinear functions, the group action on this space can be easily calculated by constructing these functions as tensor-product representations of basic representations, resulting in an explicit form of invariant feature functions. The effectiveness of our method is demonstrated on several image datasets.

## 1 Introduction

Feature coding is a method of calculating a single global feature by summarizing the statistics of the local features extracted from a single image. After obtaining the local features $\{x_n\}_{n=1}^N \in \mathbb{R}^{d_{\text{local}}}$, a nonlinear function $F$ and $F = \frac{1}{N} \sum_{n=1}^N \tilde{F}(x_n) \in \mathbb{R}^{d_{\text{global}}}$ is used as the global feature. Currently, activations of convolutional layers of pre-trained convolutional neural networks (CNNs), such as VGG-Net (Simonyan & Zisserman, 2014), are used as local features, to obtain considerable performance improvement (Sánchez et al., 2013; Wang et al., 2016). Furthermore, existing studies handle coding methods as differentiable layers and train them end-to-end to obtain high accuracy (Arandjelovic et al., 2016; Gao et al., 2016; Lin et al., 2015). Thus, feature coding is a general method for enhancing the performance of CNNs.

The invariance of images under geometric transformations is essential for image recognition because compact and discriminative features can be obtained by focusing on the information that is invariant to the transformations that preserve image content. For example, some researchers constructed CNNs with more complex invariances, such as image rotation (Cohen & Welling, 2016; 2017; Worrall et al., 2017), and obtained a model with high accuracy and reduced model parameters. Therefore, we expect to construct a feature-coding method that contains highly discriminative information per dimension and is robust to the considered transformations by exploiting the invariance information in the coding methods.

In this study, we propose a novel feature-coding method that exploits invariance. Specifically, we assume that transformations $\mathcal{T}$ that preserve image content act as a finite group consisting of orthogonal matrices on each local feature $x_n$. For example, when concatenating pixel values in the image subregion as a local feature, image flipping acts as a change in pixel values. Hence, it can be represented by a permutation matrix that is orthogonal. Also, many existing CNNs, invariant to complex transforms, implement invariance by restricting convolutional kernels such that transforms act as permutations between filter responses, which can be represented by the orthogonal matrices. Therefore, the orthogonal assumption is not a strong one.

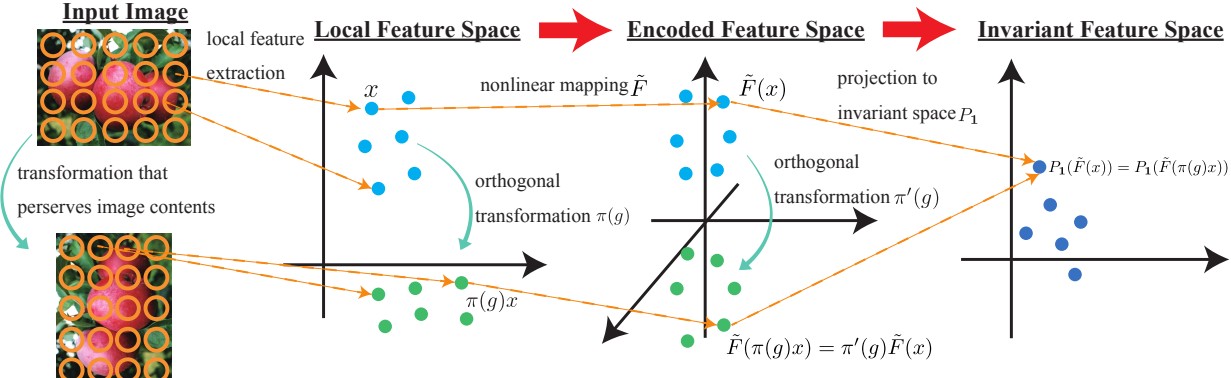

Figure 1: Overview of the proposed feature coding method. In the initially constructed global feature space, $\pi'(g)\tilde{F}(x) = \tilde{F}(\pi(g)x)$ holds for some orthogonal $\pi'$. Subsequently, the projection is applied to the trivial representation of $P_1$ to obtain the invariant global feature.

Ignoring the change in feature position, because global feature pooling is being applied, we construct a nonlinear feature coding function $F$ that exploits $\mathcal{T}$. Our first result is that when learning the linear classifier using L2-regularized convex loss minimization on the vector space, where $\mathcal{T}$ acts as an orthogonal matrix, the learned weight exists in the subspace invariant under the $\mathcal{T}$ action. From this result, we propose a guideline that first constructs a vector space in which $\mathcal{T}$ acts orthogonally on $F(x_n)$ to calculate the $\mathcal{T}$-invariant subspace.

On applying our theorem to feature coding, two problems occur when constructing the global feature. The first problem is that, in general, $\tilde{F}$ exhibits complex nonlinearity. The action of $\mathcal{T}$ on the CNN features can be easily calculated because CNNs consist of linear transformations and point-wise activation functions. This is not the case for feature-coding methods. The second is that $\tilde{F}$ has to be learned from the training data. When we encode the feature, we first apply principal component analysis (PCA) on $x_n$ to reduce the dimension. Clustering is often learned using the k-means or Gaussian mixture model (GMM) and $\tilde{F}$ is calculated from the learned model. Therefore, we must consider the effect of $\mathcal{T}$ on the learned model.

To solve these problems, we exploit two concepts of group representation theory: reducible decomposition of the representation and tensor product of two representations. The former is the decomposition of the action of $\mathcal{T}$ on $x_n$ into a direct sum of irreducible representations. This decomposition is important when constructing a dimensionality reduction method that is compatible with group actions. Subsequently, the tensor product of the representations is calculated. The tensor product is a method by which we construct a vector space where the group acts on the product of the input representations. Therefore, constructing nonlinear feature functions in which group action can be easily calculated is important.

Based on these concepts, we propose a novel feature coding method and model training method that exploit the group structure. We conducted experiments on image recognition datasets and observed an improvement in the performance and robustness to image transformations.

The contributions of this study are as follows:

- We prove that when training the linear classifier on the space where the group acts orthogonally, the learned weight lies in the invariant subspace.

- We propose a group-invariant extension to feature modeling and feature coding methods for groups acting orthogonally on local features.

- We evaluated the accuracy and invariance of our methods on image recognition datasets.

## 2 Related Work

### 2.1 Feature coding

The covariance-based approach models the distributions of local features based on Gaussian distributions and uses statistics as the global feature. For example, GLC (Nakayama et al., 2010a) uses the mean and covariance of the local descriptors as the features. The global Gaussian (Nakayama et al., 2010b) method applies an information-geometric metric to the statistical manifold of the Gaussian distribution as a similarity measure. Bilinear pooling (BP) (Lin et al., 2015) uses the mean of self-products instead of the mean and covariance, but its performance is similar. The BP is defined as $F = \text{vec}\left(\frac{1}{N}\sum_{n=1}^{N} x_n x_n^\top\right)$, where $\text{vec}(A)$ denotes the vector that stores the elements of $A$. Because of the simplicity of BP, there are various extensions, such as Lin & Maji (2017); Wang et al. (2017); Gou et al. (2018); Yu et al. (2020); Koniusz & Zhang (2021); Song et al. (2021; 2023) which demonstrate better performance than the original CNNs without a feature coding module. For example, improved bilinear pooling (iBP) uses the matrix square root as a global feature. These works mainly focus on the postprocessing of the bilinear matrix given covariance information; whereas, our motivation is to obtain the covariance information that is important to the classification. We use invariance as the criterion for feature selection.

The vector of locally aggregated descriptors (VLAD) (Jégou et al., 2010) is a $Cd_{\text{local}}$-dimensional vector that uses k-means clustering with $C$ referring to the number of clustering components and consists of the sum of the differences between each local feature and the cluster centroid $\mu_c$ to which it is assigned, expressed as $F_c = \sum_{x_n \in S_c} (x_n - \mu_c)$; $S_c$ is the set of local descriptors that are assigned to the $c$-th cluster. The vector of locally aggregated tensors (VLAT) (Picard & Gosselin, 2013) is an extension of VLAD that exploits second-order information. VLAT uses the sum of tensor products of the differences between each local descriptor and cluster centroid $\mu_c$: $F_c = \text{vec}\left(\sum_{x_n \in S_c} (x_n - \mu_c)(x_n - \mu_c)^\top - \mathcal{T}_c\right)$, where $\mathcal{T}_c$ is the mean of $(x_n - \mu_c)(x_n - \mu_c)^\top$ of all the local descriptors assigned to the $c$-th cluster. VLAT contains information similar to the full covariance of the GMM. The Fisher vector (FV) (Sánchez et al., 2013) also exploits second-order information but uses only diagonal covariance. One work has also exploited local second-order information with lower feature dimensions using local subspace clustering (Dixit & Vasconcelos, 2016).

### 2.2 Feature extraction that considers invariance

One direction for exploiting the invariance in the model structure is to calculate all transformations and subsequently apply pooling with respect to the transformation to obtain the invariant feature. TI-pooling (Laptev et al., 2016) first applies various transformations to input images, subsequently applies the same CNNs to the transformed images, and finally applies max-pooling to obtain the invariant feature. Anselmi et al. (2016) also proposed a layer that averages the activations with respect to all considered transformations. RotEqNet (Marcos et al., 2017) calculates the vector field by rotating the convolutional filters, lines them up with the activations, and subsequently applies pooling to the vector fields to obtain rotation-invariant features. Group equivariant CNNs (Cohen & Welling, 2016) construct a network based on the average of activations with respect to the transformed convolutional filters.

Another direction is to exploit the group structure of transformations and construct the group feature using group representations. Harmonic networks (Worrall et al., 2017) consider continuous image rotation in constructing a layer with spherical harmonics, which is the basis for the representation of two-dimensional rotation groups. Steerable CNNs (Cohen & Welling, 2016) construct a filter using the direct sum of the irreducible representations of the D4 group, to reduce model parameters while preserving accuracy. Weiler et al. (2018) implemented steerable CNNs as the weighted sum of predefined equivariant filters and extended it to finer rotation equivariance. Variants of steerable CNNs are summarized in (Weiler & Cesa, 2019). Jenner & Weiler (2022) constructed the equivariant layer by combining the partial differential operators. These works mainly focus on equivariant convolutional layers, whereas we mainly focus on invariant feature coding given equivariant local features. Kondor et al. (2018) also considered tensor product representation to construct DNNs that are equivariant under 3D rotation for the recognition of molecules and 3D shapes. While Kondor et al. (2018) directly uses tensor product representation as the nonlinear activation function, we use tensor product representation as the tool for extending the existing coding methods for equivariance.

Therefore, we introduce several equivariant modules, in addition to tensor product representation. The difficulty of invariant feature coding is in formulating complex nonlinear feature coding functions that are consistent with the considered transformations.

Furthermore, there exists the approach to learn invariance from training data instead of deciding the invariance beforehand. Rao & Ruderman (1998); Miao & Rao (2007) learned the group that preserves training data unsupervisedly by considering reconstruction loss between the original image and the transformed image assuming Gaussian noise. **?** used the matrix exponential and applied the matrix backpropagation to optimize the reconstruction loss. Sohl-Dickstein et al. (2010) proposed an adaptive smoothing method to obtain good local minima. Benton et al. (2020) tried to learn the neural network to be invariant under data augmentation by using consistency regularization. Lin et al. (2021) trained the auto-encoder parameterized by the Lie group. Chau et al. (2022) combined multiple 2D rotation groups with sparse coding to model the transformation group.

Another approach similar to ours is to calculate the nonlinear invariance with respect to the transformations. Reisert & Burkhardt (2006) considered the 3D rotation invariance and used the coefficients with respect to spherical harmonics as the invariant feature. Kobayashi et al. (2011) calculated a rotation-invariant feature using the norm of the Fourier coefficients. Kakarala (2012) used a bilinear feature with respect to 3D rotation group. Morère et al. (2017) proposed group-invariant statistics by integrating all the considered transformations, such as Eq. (1) for image retrieval without assuming that the transformations act linearly. Kobayashi (2017) exploited the fact that the eigenvalues of the representation of flipping are $\pm 1$ and used the absolute values of the coefficients of eigenvectors as the features. Ryu et al. (2018) used the magnitude of two-dimensional discrete Fourier transformations as the translation-invariant global feature. Compared to these approaches that manually calculate the invariants, our method can algorithmically calculate the invariants given the transformations, in fact calculating all the invariants after constructing the vector space on which the group acts. Therefore, our method exhibits both versatility and high classification performance.

In summary, our work is an extension of existing coding methods, where the effect of transformations and all the invariants are calculated algorithmically. Compared with existing invariant feature learning methods, we mainly focus on invariant feature coding from the viewpoint of group representation theory.

## 3    Overview of Group Representation Theory

In this section, we present an overview of the group representation theory that is necessary for constructing our method. The contents of this section are not original and can be found in books on group representation theory (Fulton & Harris, 2014). We present the relationship between the concepts explained in this section and the proposed method as follows:

- Group representation is the homomorphism from the group element to the matrix, meaning that the linear action is considered as the effect of transformations.

- Group representation can be decomposed into the direct sum of irreducible representations. The linear operator that commutes with group action is restricted to the direct sum of linear operators between the same irreducible representations. These results are used to formulate PCA that preserves equivariance in Section 4.2. Further, this irreducible decomposition is also used when calculating the tensor product.

- We obtain a nonlinear feature vector for the group that acts linearly by considering tensor product representation. Furthermore, the invariant vectors can be determined by exploring the subspace with respect to the trivial representation **1**, which can be obtained by Eq. (1). These results can be used to prove Theorem 1 and construct the feature coding in Section 4.3.

### 3.1    Theory

**Group representation**    When set $G$ and operation $\circ : G \times G \to G$ satisfy the following properties:

- $\circ$ is associative: $g_1, g_2, g_3 \in G$ satisfies $((g_1 \circ g_2) \circ g_3) = (g_1 \circ (g_2 \circ g_3))$

- The identity element $e \in G$ exists and satisfies $g \circ e = e \circ g = g$ for all $g \in G$

- All $g \in G$ contain the inverse $g^{-1}$ that satisfies $g \circ g^{-1} = g^{-1} \circ g = e$

the pair $\mathcal{G} = (G, \circ)$ is called a group. The axioms above are abstractions of the properties of the transformation sets. For example, a set consisting of 2D image rotations associated with the composition of transformations form a group. When the number of elements in $G$ written as $|G|$ is finite, $\mathcal{G}$ is called a finite group. As mentioned in Section 1, we consider a finite group.

Now, we consider a complex vector space $\mathbb{C}^d$ to simplify the theory; however, in our setting, the proposed global feature is real. The space of bijective operators on $\mathbb{C}^d$ can be identified with the space of $d \times d$ regular complex matrices written as $\mathrm{GL}(d, \mathbb{C})$, which is also a group with the matrix product as the operator. The homomorphism $\pi$ from $\mathcal{G}$ to $\mathrm{GL}(d, \mathbb{C})$ is the mapping $\pi : G \to \mathrm{GL}(d, \mathbb{C})$, which satisfies

- For $g_1, g_2 \in G$, $\pi(g_1) \circ \pi(g_2) = \pi(g_1 \circ g_2)$

- $\pi(e) = 1_{d \times d}$

and is called the representation of $\mathcal{G}$ on $\mathbb{C}^d$, where $1_{d \times d}$ denotes the $d$-dimensional identity matrix. The space in which the matrices act is denoted by $(\pi, \mathbb{C}^d)$. The representation that maps all $g \in G$ to $1_{d \times d}$ is called a trivial representation. The one-dimensional trivial representation is denoted as $\mathbf{1}$. When all $\pi(g)$ are unitary matrices, the representation is called a unitary representation. Furthermore, when the $\pi(g)$s are orthogonal matrices, this is called an orthogonal representation. The orthogonal representation is also a unitary representation. In this study, transformations are assumed to be orthogonal.

**Intertwining operator** For two representations $(\pi, \mathbb{C}^d)$ and $(\pi', \mathbb{C}^{d'})$, a linear operator $A : \mathbb{C}^d \to \mathbb{C}^{d'}$ is called an intertwining operator if it satisfies $\pi'(g) \circ A = A \circ \pi(g)$. This implies that $(\pi', A\mathbb{C}^d)$ is also a representation. Thus, when applying linear dimension reduction, the projection matrix must be an intertwining operator. We denote the space of the intertwining operator as $\mathrm{Hom}_G(\pi, \pi')$. When a bijective $A \in \mathrm{Hom}_G(\pi, \pi')$ exists, we write $\pi \simeq \pi'$. This implies that the two representations are virtually the same and that the only difference is the basis of the vector space.

**Irreducible representation** Given two representations $\pi$ and $\sigma$, the mapping that associates $g$ with the matrix to which we concatenate $\pi(g)$ and $\sigma(g)$ in block-diagonal form is called the direct sum representation of $\pi$ and $\sigma$, written as $\pi \oplus \sigma$. When the representation $\pi$ is equivalent to some direct sum representation, $\pi$ is called a completely reducible representation. The direct sum is the composition of the space on which the group acts independently. Therefore, a completely reducible representation can be decomposed using independent and simpler representations. When the representation is unitary, the representation that cannot be decomposed is called an irreducible representation, and all representations are equivalent to the direct sum of the irreducible representations. The irreducible representation $\tau_t$ is determined by the group structure, and $\pi$ is decomposed into $\pi \simeq n_1 \tau_1 \oplus n_2 \tau_2 \oplus ... n_T \tau_T$, where $T$ is the number of different irreducible representations and $n_t \tau_t$ is $n_t$ times direct sum of $\tau_t$. When we denote the characteristic function of $g$ as $\chi_\pi(g) = \mathrm{Tr}(\pi(g))$, we can calculate these coefficients as $n_t = \frac{1}{|G|} \sum_{g \in G} \overline{\chi_\pi(g)} \chi_{\tau_t}(g)$. Furthermore, the projection operator $P_\tau$ on $n_t \tau_t$ is calculated as $P_{\tau_t} = \dim(\tau_t) \frac{1}{|G|} \sum_{g \in G} \overline{\chi_{\tau_t}(g)} \pi(g)$. Specifically, because $\chi_{\mathbf{1}}(g) = 1$, we can calculate the projection to the trivial representation using

$$P_{\mathbf{1}} = \frac{1}{|G|} \sum_{g \in G} \pi(g). \tag{1}$$

This equation reflects the fact that the average of all $\pi(g)$ is invariant to group action. Schur's lemma indicates that $\mathrm{Hom}_G(\tau_{t_1}, \tau_{t_2}) = \{0\}$ if $t_1 \neq t_2$ and $\mathrm{Hom}_G(\tau_t, \tau_t) = \mathbb{C}A$ for some matrix $A$.

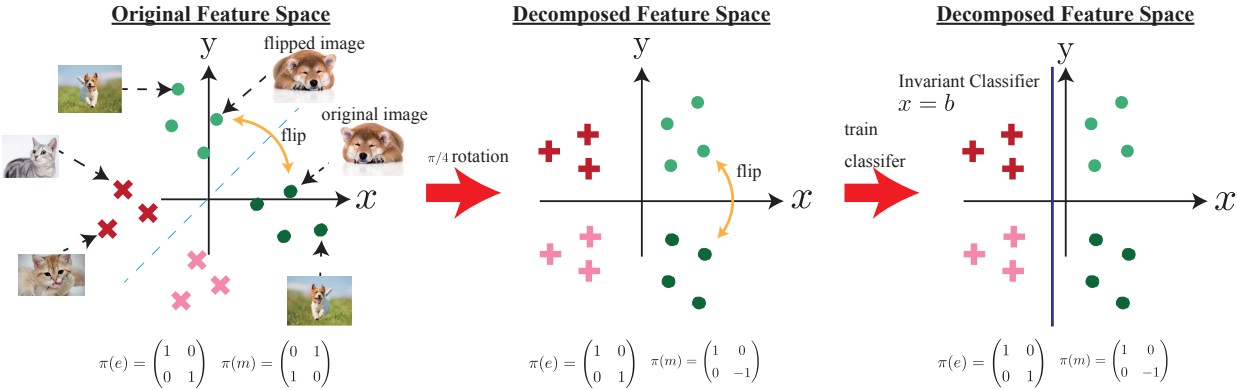

Figure 2: Example of the irreducible decomposition and learned classifier.

**Tensor product representation**   Finally, tensor product representation is important when constructing nonlinear feature functions. Given $\pi$ and $\sigma$, the mapping that associates $g$ with the matrix tensor product of $\pi(g), \sigma(g)$ is the representation of the space of the tensor product of the input spaces. We denote the tensor product representation as $\pi \otimes \sigma$. The tensor product of the unitary representation is also unitary. The important properties of the tensor representation are as follows: (i) it is distributive, where $(\pi_1 \oplus \pi_2) \otimes (\pi_3 \oplus \pi_4) = (\pi_1 \otimes \pi_3) \oplus (\pi_2 \otimes \pi_3) \oplus (\pi_1 \otimes \pi_4) \oplus (\pi_2 \otimes \pi_4)$, and (ii) $\chi_{\pi \otimes \sigma}(g) = \chi_\pi(g)\chi_\sigma(g)$. Thus, the irreducible decomposition of tensor representations can be calculated from the irreducible representations.

### 3.2   Illustrative example

The group representation theory described above is visualized in a simple setting in this section.

**Group consisting of identity mapping and image flipping.**   We consider group consists of identity mapping $e$ and horizontal image flipping $m$. Because the original image was obtained by applying image flipping twice, it follows that $e \circ e = e$, $e \circ m = m$, $m \circ e = m$, $m \circ m = e$. This definition satisfies the following three properties of a group by setting $m^{-1} = m$:

- $\circ$ is associative: $g_1, g_2, g_3 \in G$ satisfy $((g_1 \circ g_2) \circ g_3) = (g_1 \circ (g_2 \circ g_3))$.

- The identity element $e \in G$ exists and satisfies $g \circ e = e \circ g = g$ for all $g \in G$.

- All $g \in G$ contain the inverse $g^{-1}$ that satisfy $g \circ g^{-1} = g^{-1} \circ g = e$.

The irreducible representations are $\tau_1 : \tau_1(e) = 1, \tau_1(m) = 1$ and $\tau_{-1} : \tau_{-1}(e) = 1, \tau_{-1}(m) = -1$. This is proven as follows: the $\tau$s defined above satisfies the representation conditions

- For $g_1, g_2 \in G$, $\tau(g_1) \circ \tau(g_2) = \tau(g_1 \circ g_2)$.

- $\tau(e) = 1_{d \times d}$.

where 1-dimensional representations are trivially irreducible. When the dimension of the representation space is greater than 1, $\pi(e) = 1_{d \times d}$ and $\pi(m) = A \in \mathrm{GL}(d, \mathbb{C})$. We apply eigendecomposition of $A$ to obtain $A = S^{-1}\Lambda S$, where $\Lambda$ is diagonal, and we denote the $i$-th diagonal elements as $\lambda_i$. Because $\pi(e) = S^{-1}I_{d \times d}S$ and $\pi(m) = S^{-1}\Lambda S$, $\pi$ is decomposed as a direct sum representation of $\pi_i(e) = 1$ and $\pi_i(m) = \lambda_i$ for each $i$-th dimension, this representation is not irreducible. Thus, the irreducible representation must be 1-dimensional. Moreover, $\pi(m)^2 = \pi(e) = 1$. Therefore, $\pi(m)$ is 1 or -1 and $\pi$ can be deomposed into the direct sum of $\tau_1$ and $\tau_{-1}$.

**Image feature and its irreducible decomposition.** As an example of the image feature, we use concatenation of the luminosity of two horizontally adjacent pixels as a local feature and apply average pooling to get a global feature. Thus, the feature dimension is 2. Since image flipping changes the order of the pixels, it permutates the first and second elements in feature space. Therefore, the group acts as $\pi(e) = \begin{pmatrix} 1 & 0 \\ 0 & 1 \end{pmatrix}$, $\pi(m) = \begin{pmatrix} 0 & 1 \\ 1 & 0 \end{pmatrix}$, as plotted in the left figure of Figure 2. By applying orthogonal matrices calculated by eigendecomposition, the feature space written in the center of Figure 2 is obtained. In this space, the group acts as $\pi(e) = \begin{pmatrix} 1 & 0 \\ 0 & 1 \end{pmatrix}$, $\pi(m) = \begin{pmatrix} 1 & 0 \\ 0 & -1 \end{pmatrix}$. Thus, this is the irreducible decomposition of $\pi$ into $\tau_1 \oplus \tau_{-1}$ defined by the **Irreducible representation** paragraph in the previous subsection. **Note that in the general case, each representation matrix is block-diagonalized instead of diagonalized, and each diagonal block becomes more complex, like those in Tables 1 and 7.**

**Tensor product representation.** Subsequently, we consider the "product" of feature spaces to get non-linear features. Two feature spaces $(h_a^1, h_a^2)$ and $(h_b^1, h_b^2)$ with $\pi$ acting as $\pi(e) = \begin{pmatrix} 1 & 0 \\ 0 & 1 \end{pmatrix}$, $\pi(m) = \begin{pmatrix} 0 & 1 \\ 1 & 0 \end{pmatrix}$ on both spaces are considered. Any input spaces can be used whenever these spaces and $\pi$ satisfy the above condition. For example, we use the same feature space for $(h_a^1, h_a^2)$ and $(h_b^1, h_b^2)$ to obtain bilinear pooling. The tensor product of these spaces becomes a 4-dimensional vector space consisting of $h_a^1 h_b^1$, $h_a^1 h_b^2$, $h_a^2 h_b^1$ and $h_a^2 h_b^2$. Since $m$ permutates $h_a^1$ and $h_a^2$, $h_b^1$ and $h_b^2$ simultaneously, $m$ acts as a permutator of $h_a^1 h_b^1$ and $h_a^2 h_b^2$, $h_a^1 h_b^2$ and $h_a^2 h_b^1$ simultaneously. Thus, $\pi(m)$ is written as $\pi(m) = \begin{pmatrix} 0 & 0 & 0 & 1 \\ 0 & 0 & 1 & 0 \\ 0 & 1 & 0 & 0 \\ 1 & 0 & 0 & 0 \end{pmatrix}$. Moreover, tensor product space is also the group representation space. This space can also be decomposed into irreducible representations, such as $\pi(e) = \begin{pmatrix} 1 & 0 & 0 & 0 \\ 0 & 1 & 0 & 0 \\ 0 & 0 & 1 & 0 \\ 0 & 0 & 0 & 1 \end{pmatrix}$ and $\pi(m) = \begin{pmatrix} 1 & 0 & 0 & 0 \\ 0 & 1 & 0 & 0 \\ 0 & 0 & -1 & 0 \\ 0 & 0 & 0 & -1 \end{pmatrix}$.

**Intertwining operator** Given $(\pi, \mathbb{C}^4)$ and $(\pi', \mathbb{C}^2)$ where $\pi(e) = \begin{pmatrix} 1 & 0 & 0 & 0 \\ 0 & 1 & 0 & 0 \\ 0 & 0 & 1 & 0 \\ 0 & 0 & 0 & 1 \end{pmatrix}$ and $\pi(m) = \begin{pmatrix} 1 & 0 & 0 & 0 \\ 0 & 1 & 0 & 0 \\ 0 & 0 & -1 & 0 \\ 0 & 0 & 0 & -1 \end{pmatrix}$ and $\pi'(e) = \begin{pmatrix} 1 & 0 \\ 0 & 1 \end{pmatrix}$, $\pi'(m) = \begin{pmatrix} 1 & 0 \\ 0 & -1 \end{pmatrix}$, intertwining operator from $\pi$ to $\pi'$ maps $\tau_1$ elements to $\tau_1$ elements and $\tau_{-1}$ elements to $\tau_{-1}$ elements and therefore is represented by the form $\begin{pmatrix} a_{11} & a_{12} & 0 & 0 \\ 0 & 0 & a_{23} & a_{24} \end{pmatrix}$ where $a_{ij}$s are complex.

## 4 Invariant Tensor Feature Coding

In this section, the proposed method is explained. Our goal was to construct an effective feature function $F = \frac{1}{N} \sum_{n=1}^N \tilde{F}(x_n)$, where the transformations that preserve the image content act as a finite group of orthogonal matrices on $x_n$. Hence, we prove a theorem that reveals the condition of the invariant feature with sufficient discriminative information in Section 4.1. Subsequently, the feature modeling method necessary for constructing the coding in Section 4.2, is explained. Our proposed invariant feature function is described in Section 4.3. In Section 4.4, we discuss the limitations of this theorem and extend it under more general assumptions. In Section 4.5, the effectiveness of the proposed method is discussed in an end-to-end setting.

### 4.1 Guideline for the invariant features

#### 4.1.1 Theory

First, to determine what an effective feature is, we prove the following theorem:

**Theorem 1.** *We denote by $\mathcal{C}$ the set of image categories, $W \in \mathbb{R}^{d \times |\mathcal{C}|}$ the linear classification weight whose c-th column vector corresponds to the weight for c-th category, $\|\cdot\|_F$ the Frobenius norm of the matrix and $l : \mathbb{R}^{|\mathcal{C}|} \times \mathcal{C} \to \mathbb{R}$ be the loss function convex with respect to the first argument. Let the finite group $\mathcal{G}$ act as an orthogonal representation $\pi$ on $\mathbb{R}^d$ and preserve the distribution of the training data $\{(v_m, y_m)\}_{m=1}^M \in \mathbb{R}^d \times \mathcal{C}$, which implies that $\{(v_m, y_m)\}_{m=1}^M \in \mathbb{R}^d \times \mathcal{C}$ exhibits the same distribution as $\{(\pi(g)v_m, y_m)\}_{m=1}^M \in \mathbb{R}^d \times \mathcal{C}$ for any g. The solution of the L2-regularized convex loss minimization*

$$\underset{W \in \mathbb{R}^{d \times |\mathcal{C}|}}{\arg\min} \frac{\lambda}{2} \|W\|_F^2 + \frac{1}{M} \sum_{m=1}^M l(W^\top v_m, y_m) \tag{2}$$

*is $\mathcal{G}$-invariant, implying that $\pi(g)W = W$ for any g and therefore $P_1 W = W$.*

The $\mathcal{G}$-invariance of the training data corresponds to the fact that $g$ does not change the image content. From another viewpoint, this corresponds to data augmentation that uses the transformed images as additional training data. For example, given the original dataset $\{(u_o, y_o)\}_{o=1}^O$ and a constructed augmented dataset $\{(v_m, y_m)\}_{m=1}^M$ as $\{(\pi(g)u_o, y_o)\}_{o=1, g \in G}^O$, this augmented dataset is invariant under the action of $\pi(g)$ for any $g$. We intend to use this theorem by regarding the global feature $F$ as $v_m$, but it is applicable whenever the assumption holds. The proof is as follows.

*Proof.* The non-trivial unitary representation $\tau$ satisfies $\sum_{g \in G} \tau(g) = 0$. This is because if we assume that $\sum_{g \in G} \tau(g) = A \neq 0$, there exists $v$ that satisfies $Av \neq 0$, and $\mathbb{C}Av$ is a one-dimensional $\mathcal{G}$-invariant subspace. It violates the irreducibility of $\pi$. As $(\pi, \mathbb{R}^d)$ is a unitary representation, it is completely reducible. We denote the $n_t \tau_t$ elements of $W$ and $v_m$ as $W^{(t)}$ and $v_m^{(t)}$, respectively, where the decomposition of $W$ is applied column-wise. It follows that $W = \sum_{t=1}^T W^{(t)}$ and $v_m = \sum_{t=1}^T v_m^{(t)}$. Furthermore, since $W^{(t)}$s and $v_m^{(t)}$s for different $t$s lie in a different subspace with respect to the irreducible decomposition, they are orthogonal among different $t$s. It follows that,

$$\frac{1}{M} \sum_{m=1}^M l(W^\top v_m, y_m)$$

$$= \frac{1}{M} \sum_{m=1}^M l\left(\mathrm{Re}\left(\sum_{t=1}^T \left(W^{(t)}\right)^\top v_m^{(t)}\right), y_m\right)$$

$$= \frac{1}{M|G|} \sum_{m=1}^M \sum_{g \in G} l\left(\mathrm{Re}\left(\sum_{t=1}^T \left(W^{(t)}\right)^\top \tau_t(g^{-1})v_m^{(t)}\right), y_m\right)$$

$$= \frac{1}{M|G|} \sum_{m=1}^M \sum_{g \in G} l\left(\mathrm{Re}\left(\sum_{t=1}^T \left(\tau_t(g)W^{(t)}\right)^\top v_m^{(t)}\right), y_m\right)$$

$$\geq \frac{1}{M} \sum_{m=1}^M l\left(\sum_{t=1}^T \mathrm{Re}\left(\left(\frac{1}{|G|} \sum_{g \in G} \tau_t(g)W^{(t)}\right)^\top v_m^{(t)}\right), y_m\right)$$

$$= \frac{1}{M} \sum_{m=1}^M l\left(\left(W^{(1)}\right)^\top v_m^{(1)}, y_m\right), \tag{3}$$

where Re is the real part of a complex number and $W^{(1)}$ and $v_m^1$ are the trivial representation elements under the irreducible decomposition of $W$ and $v_m$ respectively, which can be obtained by applying the

projection matrix calculated by Eq. (1). The first equation originates from the orthogonality of $W^{(t)}$s and $v_m^{(t)}$s among different $t$s; the second equation comes from $\mathcal{G}$-invariance of the training data such that $\sum_{m=1}^M f(v_m, y_m) = \sum_{m=1}^M f(\pi(g)v_m, y_m)$ holds for any $f, g$ from the assumption; and the third comes from the unitarity of $\tau_t(g)$. The inequality comes from the convexity of $l$, additivity of Re, and inner products. The final equality comes from the fact that the average of $\tau_t(g)$ is equal to 0 for nontrivial $\tau_t$; $W^{(1)}$ and $v_m^1$ are real. Therefore, this proof exploits the properties of convex functions and group representation theory.

Combined with the fact that $\|W\|_F^2 \geq \|W^{(1)}\|_F^2$, the loss value of $W$ is larger than that of $W^{(1)}$. Therefore, the solution is $\mathcal{G}$-invariant. $\qquad\qquad\square$

We can also prove the same results by focusing on the subgradient, which is described in the Supplementary Material section. Facts from representation theory are necessary for both cases.

This theorem indicates that the complexity of the problem can be reduced by imposing invariance. Because the generalization error increases with complexity, this theorem explains one reason that invariance contributes to the good test accuracy. Sokolic et al. (2017) analyzed the generalization error in a similar setting, using a covering number. This work calculated the upper bound of complexity; whereas, we focus on the linear classifier, obtain the explicit complexity, and calculate the learned classifier. Furthermore, even when the obtained classifier is the same as that trained by the augmented samples, the invariant features are more suitable for training owing to the reduced number of feature dimension and training samples, which may affect the performance of the learned feature. Moreover, we can apply a larger size of code words for the clustering-based methods when we match the global feature dimension. Hence, our goal is to **construct a feature that is invariant in the vector space where the group acts orthogonally.**

### 4.1.2 Illustrative example

Theory 1 is illustrated in the setting of Section 3.2. In this decomposed space, when the classifier is trained by considering both original images and flipped images, the classification boundary learned with L2-regularized convex loss minimization acquires the form of $x = b$, as plotted in the right figure of Figure 2. Thus, we can discard $y$-elements of features and obtain a compact feature vector. This result is validated as follows: when the feature in the decomposed space that corresponds to the original image is denoted as $(v^1, v^2)$, the feature corresponding to the flipped image becomes $(v^1, -v^2)$ because $\pi(m) = \begin{pmatrix} 1 & 0 \\ 0 & -1 \end{pmatrix}$. Expressing the linear classifier as $w^1 v^1 + w^2 v^2 + b$, the loss for these two images can be written as $l(w^1 v^1 + w^2 v^2 + b) + l(w^1 v^1 - w^2 v^2 + b)$. From Jensen's inequality, this is lower-bounded by $2l(w^1 v^1 + b)$. This means that the loss is minimized when $w^2 = 0$, resulting in the classification boundary $w^1 v^1 + b = 0$. $w^1$ is invariant under the action of $\pi(e)$ and $\pi(m)$.

### 4.2 Invariant feature modeling

First, a feature-modeling method is constructed for the calculation of invariant feature coding.

**Invariant PCA** PCA attempts to determine the subspace that maximizes the sum of the variances of the projected vectors. The original PCA is a solution to $\max_{U^\top U = I} \mathrm{Tr}\left(U^\top \frac{1}{N} \sum_{n=1}^N (x_n - \mu)(x_n - \mu)^\top U\right)$, where $\mu = \frac{1}{N} \sum_{n=1}^N x_n$. The solution is a matrix consisting of the eigenvectors of $\frac{1}{N} \sum_{n=1}^N (x_n - \mu)(x_n - \mu)^\top$ that correspond to the top eigenvectors.

Our method can only utilize the projected low-dimensional vector if it also lies in the representation space of the considered group. Therefore, in addition to the original constraint $U^\top U = I$, we assume that $U$ is an intertwining operator in the projected space. From Schur's lemma described in Section 3, the $U$ that satisfies these conditions is the matrix obtained using the projection operator $P_{\tau_t}$ with the dimensionality reduced to $n_t \tau_t$. Furthermore, when $n_t \tau_t = \bigoplus_{o=1}^{n_t} \tau_{t,o}$ and the $\tau_{t,o}$-th element of $x_n$ as $x_n^{(t,o)}$, the dimensionality reduction within $n_t \tau_t$ is in the form of $x_n^{(t)} \to \sum_{o=1}^{n_t} u^{(t,o)} x_n^{(t,o)}$ for $u^{(t,o)} \in \mathbb{C}$ because of Schur's lemma. Hence, our basic strategy is to first calculate $u^{(t,o)}$s that maximize the variance with the orthonormality preserved and subsequently choose $t$ for larger variances per dimension. In fact, $u^{(t,o)}$ can be calculated using PCA with

---

**Algorithm 1** Calculation of Invariant PCA

---

**Input:** $\{x_n\}_{n=1}^N \in \mathbb{R}^d, \mathcal{G}, d_{\text{proj}}$
**Output:** $U \in \mathbb{R}^{d \times d_{\text{proj}}}$ which is intertwining and orthonormal
   **for** $t = 1$ to $T$ **do**
      **for** $o_1, o_2 = 1$ to $n_t$ **do**
         $\Sigma^{(t)}{}_{o_1,o_2} \leftarrow \frac{1}{N} \sum_{n=1}^N \left\langle x_n^{(t,o_1)} - \mu^{(t,o_1)}, x_n^{(t,o_2)} - \mu^{(t,o_2)} \right\rangle_{\mathbb{R}}$
      **end for**
      $(\lambda_p^{(t)}, u_p^{(t)}) \leftarrow$ eigendecomposition of $\Sigma^{(t)}$.
   **end for**
   $U =$ empty matrix
   **while** size of $U$ is smaller than $d \times d_{\text{proj}}$ **do**
      $U \leftarrow$ concat of $U$ and $\left( u_p^{(t)} \otimes I_{d_{\tau_t}} \right) \circ P_{\tau_t}$ for non-used $p, t$ with maximum $\lambda_p^{(t)}/d_{\tau_t}$
   **end while**

---

**Algorithm 2** Calculation of invariant k-means

---

**Input:** $\{x_n\}_{n=1}^N \in \mathbb{R}^d, \mathcal{G}, C$
**Output:** $\mu_{g,c}$ for $g \in G, c \in \{1, ..., C\}$
   randomly initialize $\mu_{e,c}$ for $c \in \{1, ..., C\}$
   **for** it $= 1$ to maxiter **do**
      $\mu_{g,c} \leftarrow \pi(g)\mu_{e,c}$ for $g \in G, c \in \{1, ..., C\}$
      $S_{g,c} \leftarrow \{n | (g,c) = \arg \min_{(g,c)} \|x_n - \mu_{g,c}\|\}$
      $\mu_{e,c} \leftarrow \frac{1}{\sum_{g \in G} |S_{g,c}|} \sum_{g \in G} \sum_{n \in S_{g,c}} \pi(g)^{-1} x_n$ for $c \in \{1, ..., C\}$.
   **end for**

---

the sum of covariances between each dimensional element of $x_n^{(t,o)}$. This algorithm is presented in Algorithm 1. Because the projected vector must be real, additional care is required when certain elements of $\tau_t$ are complex. The modification for this case are described in the Supplementary Materials section. In the experiment, we use groups in which all irreducible representations can be real; thus, $\tau_t$ can be used directly.

As for computational complexity, standard PCA requires square order of the input feature dimension times the number of input features plus cubic order of the input feature dimension and therefore $O\left( N \left( \sum_{t=1}^T n_t \dim(\tau_t) \right)^2 + \left( \sum_{t=1}^T n_t \dim(\tau_t) \right)^3 \right)$. On the other hand, the proposed invariant PCA requires the computation of PCA for each irreducible representation and therefore $O\left( \sum_{t=1}^T \left( N n_t^2 \dim(\tau_t) + n_t^3 \right) \right)$.

Note that PCA considers the 2D rotation using Fourier transformation and has been applied to cryo-EM images (Zhao & Singer, 2013; Vonesch et al., 2015; 2013; Zhao et al., 2016). The proposed PCA can be regarded as an extension of the general noncommutative group. Other studies (Zhao & Singer, 2013; Vonesch et al., 2015; 2013; Zhao et al., 2016) apply PCA for direct image modeling, whereas we regard our proposed PCA as a preprocessing method that preserves the equivariance property for successive feature coding.

**Invariant k-means** The original k-means algorithm is calculated as $\min_\mu \sum_{n=1}^N \min_{c=1}^C \|x_n - \mu_c\|^2$. To ensure that $\mathcal{G}$ acts orthogonally on the learned model, we simply retain the cluster centroids by applying all transformations to the original centroid. This implies that we learn $\mu_{g,c}$ for $g \in G, c \in \{1, ..., C\}$ such that $\mu_{g,c}$ satisfies $\pi(g_1)^{-1}\mu_{g_1,c} = \pi(g_2)^{-1}\mu_{g_2,c}$ for all $g_1, g_2, c$. This algorithm is listed in Algorithm 2. As for computational complexity, the proposed k-means requires $|G|$ times as large computation complexity as the original k-means since the size of the code words becomes $|G|$ times as large.

### 4.3 Invariant feature coding

Subsequently, we construct a feature coding function $F$ as a $\mathcal{G}$-invariant vector in the space where $\mathcal{G}$ acts orthogonally. First, the spaces of the function $x_n$ that $\mathcal{G}$ act orthogonally are calculated as the basis spaces for constructing more complex representation spaces using the tensor product. The first basis space is that of $x_n$. The second is the $C|G|$-dimensional 1-of-k vector that we assign $x_n$ to the nearest $\mu_{g,c}$. When we apply $\pi(g)$ to $x_n$, the nearest $\mu$ is $\mu_{g,c}$ which corresponds to the vector to which we apply $\pi(g)$ on the nearest $\mu$ to $x_n$. Therefore, $g$ acts as a permutation. We denote this representation by $1_\mu(g)$.

**Invariant BP**  Because BP is written as the tensor product of $x_n$, we use

$$F = \text{vec} \left( \frac{1}{N} \sum_{n=1}^{N} P_{\mathbf{1}} \left( x_n \otimes x_n \right) \right), \tag{4}$$

as the invariant global feature. Although $P_{\mathbf{1}}$ is a projection onto the trivial representation defined by Eq. (1), we can calculate this feature from the irreducible decomposition of $x_n$ and the irreducible decomposition of the tensor products. Normalization can also be applied to the invariant covariance with respect to each $P_{\mathbf{1}} \left( \tau_{t,o} \otimes \tau_{t,o} \right)$-th element to obtain a variant of BP, such as iBP (Lin & Maji, 2017). Because elements that are not invariant are discarded, the feature dimensions become smaller.

**Invariant VLAD**  Subsequently, we propose an invariant version of VLAD. The first difficulty is that VLAD is not in the form of a tensor product, and second, the effect of transformation on the nearest code word is not consistent. We solved the first difficulty by considering VLAD as the tensor product of the local features and 1-of-k representation, and the second difficulty was solved by using the invariant k-means proposed in Section 4.2 to learn the code word.

The expression for the invariant VLAD is as follows:

$$F = \text{vec} \left( \frac{1}{N} \sum_{n=1}^{N} P_{\mathbf{1}} \left( 1_\mu(x_n) \otimes (x_n - \mu_c) \right) \right), \tag{5}$$

where $\mu_c$ denotes the nearest centroid to $x_n$. Because $1_\mu(x_n)$ is a vector in which only the element corresponding to the nearest $\mu$ is 1, the tensor product becomes the vector in which the elements corresponding to the nearest $\mu$ are $x_n - \mu_c$, which is the same as the original VLAD. Because both $1_\mu(x_n)$ and $x_n - \mu_c$ are orthogonal representation spaces, this space is also an orthogonal representation space. Although the size of the codebook becomes $|G|$times larger, the dimensions of the global feature are not as large because we used the invariant vector.

**Illustrative example of invariant k-means and VLAD**  As shown in Figure 3, when the learned codebook is $\mu_i$ for $i = 1, 2, 3$, $x_1$ and $x_2$ are both assigned to $\mu_1$. When these features are flipped, the flipped $x_1$ is assigned to $\mu_3$, whereas the flipped $x_2$ is assigned to $\mu_2$. Thus, image flipping does not act consistently on the assignment vector $1_\mu(x_n)$. As plotted in Figure 4, when the codebook is learned such that there exists $\mu$ with $y$-element flipped for each code word, the group acts consistently on $1_\mu(x_n)$. This is because whenever $x_n$ is assigned to $\mu_{e,i}$, the flipped $x_n$ is assigned to $\mu_{m,i}$. In addition, the flipped $x_n$ is assigned to $\mu_{e,i}$ when $x_n$ is assigned to $\mu_{m,i}$. Therefore, the group acts orthogonally on $1_\mu(x_n)$.

**Invariant VLAT**  Finally, we propose the invariant VLAT that incorporates local second-order statistics. This feature can be calculated by combining the two aforementioned features.

$$F = \text{vec} \left( \frac{1}{N} \sum_{n=1}^{N} P_{\mathbf{1}} \left( 1_\mu(x_n) \otimes ((x_n - \mu_c) \otimes (x_n - \mu_c) - \mathcal{T}_c) \right) \right). \tag{6}$$

The dimension of the invariant VLAT with $C|G|$ components is the same as that of VLAT with $C$ components.

Thus, we can model complex nonlinear statistics and calculate the invariants using the tensor product representation when using local polynomial statistics as a global feature.

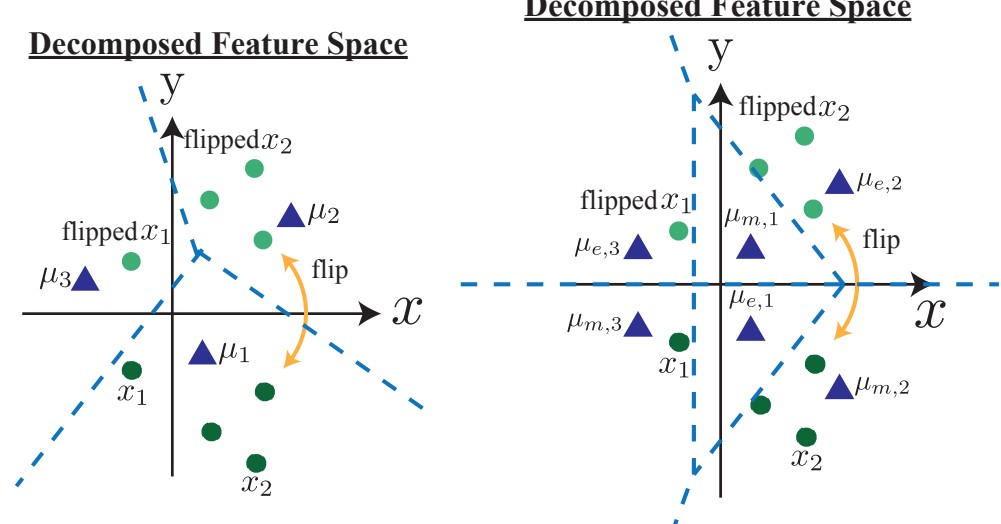

Figure 3: VLAD with code words learned by k-means.

Figure 4: Invariant VLAD with code words learned by proposed k-means.

Table 1: Irreducible representations of the D4 group.

| Rep. | $e$ | $r$ | $r^2$ | $r^3$ | $m$ | $mr$ | $mr^2$ | $mr^3$ |
|---|---|---|---|---|---|---|---|---|
| $\tau_{1,1}$ | 1 | 1 | 1 | 1 | 1 | 1 | 1 | 1 |
| $\tau_{1,-1}$ | 1 | 1 | 1 | 1 | -1 | -1 | -1 | -1 |
| $\tau_{-1,1}$ | 1 | -1 | 1 | -1 | 1 | -1 | 1 | -1 |
| $\tau_{-1,-1}$ | 1 | -1 | 1 | -1 | -1 | 1 | -1 | 1 |
| $\tau_2$ | $\begin{bmatrix} 1 & 0 \\ 0 & 1 \end{bmatrix}$ | $\begin{bmatrix} 0 & -1 \\ 1 & 0 \end{bmatrix}$ | $\begin{bmatrix} -1 & 0 \\ 0 & -1 \end{bmatrix}$ | $\begin{bmatrix} 0 & 1 \\ -1 & 0 \end{bmatrix}$ | $\begin{bmatrix} -1 & 0 \\ 0 & 1 \end{bmatrix}$ | $\begin{bmatrix} 0 & 1 \\ 1 & 0 \end{bmatrix}$ | $\begin{bmatrix} 1 & 0 \\ 0 & -1 \end{bmatrix}$ | $\begin{bmatrix} 0 & -1 \\ -1 & 0 \end{bmatrix}$ |

### 4.4 Limitation and extension

Our theorem and methods depend on two assumptions regarding the transformations: finiteness and orthogonality. At first glance, these assumptions may seem strong. However, the outputs of a wide variety of equivariant CNNs satisfy these requirements. This is because the invariance with respect to infinite transformations can often be approximated by finite invariance. For example, the continuous rotation invariance is approximated with small discrete rotations (Marcos et al., 2017; Weiler et al., 2018). Moreover, invariance is often implemented by CNNs with special restrictions on convolutional filters (e.g. filter "A" is filter "B" rotated by some $\theta$). In this case, the transformations act as permutations between the filter responses, which are orthogonal. Therefore, these assumptions are not restrictive. Furthermore, we can extend Theorem 1 to a more general assumption by exploiting a more advanced group representation theory. This case is discussed in the Supplementary Materials section.

### 4.5 Effectiveness of the end-to-end setting

Although Theorem 1 assumes that the input features are fixed, the effectiveness of the proposed invariant feature can also be demonstrated when the local features are learned in an end-to-end manner. We write the entire classification model as $W^\top F_{\theta_c} f_{\theta_l}(x)$, where $F_{\theta_c}$ and $f_{theta_l}$ denote the learnable feature coding function and feature extractor, respectively. We assume that these functions preserve equivariance, meaning that the transformations act orthogonally on $F_{\theta_c} f_{\theta_l}(x)$. This assumption is satisfied when equivariant CNNs are used as the local feature extractor along with the proposed invariant feature coding functions. Then, the

Table 2: Irreducible representations of tensor products of irreducible representations of the D4 group.

| | $\tau_{1,1}$ | $\tau_{1,-1}$ | $\tau_{-1,1}$ | $\tau_{-1,-1}$ | $\tau_2$ |
|---|---|---|---|---|---|
| $\tau_{1,1}$ | $\tau_{1,1}$ | $\tau_{1,-1}$ | $\tau_{-1,1}$ | $\tau_{-1,-1}$ | $\tau_2$ |
| $\tau_{1,-1}$ | $\tau_{1,-1}$ | $\tau_{1,1}$ | $\tau_{-1,-1}$ | $\tau_{-1,1}$ | $\tau_2$ |
| $\tau_{-1,1}$ | $\tau_{-1,-1}$ | $\tau_{-1,1}$ | $\tau_{1,-1}$ | $\tau_{1,1}$ | $\tau_2$ |
| $\tau_{-1,-1}$ | $\tau_{-1,-1}$ | $\tau_{-1,1}$ | $\tau_{1,-1}$ | $\tau_{1,1}$ | $\tau_2$ |
| $\tau_2$ | $\tau_2$ | $\tau_2$ | $\tau_2$ | $\tau_2$ | $\tau_{1,1} \oplus \tau_{1,-1} \oplus \tau_{-1,1} \oplus \tau_{-1,-1}$ |

Table 3: Comparison of accuracy using fixed features.

| Methods | Dim. | FMD | DTD | UIUC | CUB | Cars |
|---|---|---|---|---|---|---|
| BP | 525k | $81.28 \pm 1.54$ | $75.89 \pm 0.72$ | $80.83 \pm 2.35$ | 77.48 | 86.22 |
| iBP | 525k | $81.38 \pm 1.38$ | $75.88 \pm 0.62$ | $81.94 \pm 3.19$ | 75.90 | 86.74 |
| VLAD | 525k | $80.38 \pm 1.61$ | $75.12 \pm 0.77$ | $80.37 \pm 3.23$ | 72.94 | 86.10 |
| VLAT | 525k | $79.98 \pm 1.66$ | $76.24 \pm 0.66$ | $80.46 \pm 2.60$ | 76.62 | 87.12 |
| FV | 525k | $78.18 \pm 1.66$ | $75.56 \pm 0.80$ | $79.35 \pm 1.52$ | 66.59 | 81.70 |
| Inv BP (ours) | 82k | $83.34 \pm 1.48$ | $77.19 \pm 0.60$ | $81.48 \pm 1.75$ | 81.79 | 87.45 |
| Inv iBP (ours) | 82k | $\mathbf{83.46 \pm 1.27}$ | $\mathbf{77.96 \pm 0.46}$ | $\mathbf{83.33 \pm 1.80}$ | 82.12 | **88.80** |
| Inv VLAD (ours) | 525k | $82.42 \pm 1.74$ | $76.53 \pm 0.56$ | $81.94 \pm 1.40$ | 80.97 | 88.54 |
| Inv VLAT (ours) | 525k | $81.88 \pm 1.34$ | $77.45 \pm 0.55$ | $82.13 \pm 1.64$ | **83.25** | 88.59 |

learning problem can be formulated as

$$\arg \min_{W \in \mathbb{R}^{d \times |\mathcal{C}|}, \theta_c, \theta_l} \frac{\lambda}{2} \|W\|_F^2 + \frac{1}{M} \sum_{m=1}^{M} l(W^\top F_{\theta_c} f_{\theta_l}(x_m), y_m). \tag{7}$$

For any fixed $\theta$, we can apply Theorem 1 by considering $v_m = F_{\theta_c} f_{\theta_l}(x_m)$. Therefore, the invariant feature is effective, even in an end-to-end setting.

## 5 Experiment

In this section, the accuracy and invariance of the proposed method are evaluated using the pretrained features in Section 5.1. In Section 5.2, we evaluate the method on an end-to-end case.

### 5.1 Experiment with fixed local features

In this subsection, our methods are evaluated on image recognition datasets using pretrained CNN local features. Note that the local features were fixed to compare only the performance of coding methods; thus, the overall scores are lower than those of the existing end-to-end methods.

These methods were evaluated using the Flickr Material Dataset (FMD) (Sharan et al., 2013), describable texture datasets (DTD) (Cimpoi et al., 2014), UIUC material dataset (UIUC) (Liao et al., 2013), Caltech-UCSD Birds (CUB) (Welinder et al., 2010)) and Stanford Cars (Cars) (Krause et al., 2013). FMD contains 10 material categories with 1,000 images; DTD contains 47 texture categories with 5,640 images; UIUC contains 18 categories with 216 images; CUB contains 200 bird categories with 11,788 images; and Cars consists of 196 car categories with 16,185 images. We used given training test splits for DTD, CUB, and Cars. We randomly split 10 times such that the sizes of the training and testing data would be the same for each category for FMD and UIUC.

### 5.1.1 Results on D4 Group

First, we applied our method to the D4 group used in Cohen & Welling (2017), which contains rich information and is easy to calculate. The D4 group consists of $\pi/2$ rotation $r$ and image flipping $m$ with $|G| = 8$.

Table 4: Comparison of accuracy using fixed features on the augmented test data.

| Methods | Dim. | FMD | DTD | UIUC | CUB | Cars |
|---|---|---|---|---|---|---|
| BP | 525k | $78.17 \pm 1.32$ | $70.90 \pm 0.64$ | $69.12 \pm 1.53$ | 37.35 | 27.45 |
| iBP | 525k | $78.28 \pm 1.17$ | $71.00 \pm 0.64$ | $69.39 \pm 1.91$ | 38.32 | 28.45 |
| VLAD | 525k | $77.20 \pm 1.16$ | $70.62 \pm 0.77$ | $67.95 \pm 2.18$ | 36.18 | 29.78 |
| VLAT | 525k | $77.03 \pm 1.14$ | $71.45 \pm 0.65$ | $69.87 \pm 1.53$ | 41.01 | 27.27 |
| FV | 525k | $75.28 \pm 1.41$ | $70.78 \pm 0.76$ | $65.89 \pm 1.06$ | 29.36 | 27.37 |
| Inv BP (ours) | 82k | $83.05 \pm 1.43$ | $77.09 \pm 0.61$ | $81.48 \pm 1.75$ | 81.62 | 87.50 |
| Inv iBP (ours) | 82k | $\mathbf{83.24 \pm 1.19}$ | $\mathbf{77.83 \pm 0.47}$ | $\mathbf{83.38 \pm 1.81}$ | 81.98 | $\mathbf{88.80}$ |
| Inv VLAD (ours) | 525k | $82.14 \pm 1.67$ | $76.38 \pm 0.50$ | $81.68 \pm 1.32$ | 80.75 | 88.54 |
| Inv VLAT (ours) | 525k | $81.77 \pm 1.31$ | $77.36 \pm 0.60$ | $82.13 \pm 1.64$ | $\mathbf{83.01}$ | 88.62 |

Table 5: Comparison of accuracy using invariant feature modeling and existing feature coding methods.

| Methods | Dim. | FMD | DTD | UIUC | CUB | Cars |
|---|---|---|---|---|---|---|
| BP | 525k | $81.20 \pm 1.60$ | $75.09 \pm 0.56$ | $80.19 \pm 3.00$ | 78.24 | 85.67 |
| iBP | 525k | $81.20 \pm 1.60$ | $75.05 \pm 0.63$ | $81.20 \pm 3.35$ | 76.36 | 86.00 |
| VLAD | 525k | $80.50 \pm 1.95$ | $75.20 \pm 0.86$ | $78.80 \pm 2.29$ | 71.61 | 85.72 |
| VLAT | 525k | $79.98 \pm 1.62$ | $75.66 \pm 0.82$ | $81.02 \pm 1.86$ | 78.29 | 84.77 |
| FV | 525k | $78.50 \pm 1.65$ | $75.63 \pm 0.72$ | $79.91 \pm 1.80$ | 67.47 | 82.47 |

The irreducible representations and decomposition of the tensor products are summarized in Tables 1 and 2, respectively.

Because D4 is not orthogonal to the output of the standard CNNs, we pretrained the group equivariant CNNs and used the last convolutional activation as the local feature extractor. The group equivariant CNN is the model in which we preserve the $\mathcal{G}$ action using $|G|$ times the number of filters $\pi(g)$ applied on the original filters and use the average with respect to $g$ for the convolution. When the feature is of $d_{\mathrm{CNN}} \times |G|$ dimension, it can be regarded as $d_{\mathrm{CNN}}$ times the direct sum of the eight-dimensional orthogonal representation space because D4 acts as a permutation. The representation was decomposed as follows: $\pi_{\mathrm{CNN}} = d_{\mathrm{CNN}}\tau_{1,1} \oplus d_{\mathrm{CNN}}\tau_{1,-1} \oplus d_{\mathrm{CNN}}.\tau_{-1,1} \oplus d_{\mathrm{CNN}}\tau_{-1,-1} \oplus 2d_{\mathrm{CNN}}\tau_2$.

The group equivariant CNNs with the VGG19 architecture were pretrained with convolutional filter sizes of $23, 45, 91, 181, 181$ instead of $64, 128, 256, 512, 512$ as the local feature extractor. Furthermore, we added batch normalization layers for each convolution layer to accelerate the training. The model was trained using the ILSVRC2012 dataset (Deng et al., 2009). A standard data augmentation strategy was applied, and the same learning settings as the original VGG-Net were used. The pretraining code was implemented using Pytorch (Paszke et al., 2019) with the group equivariant convolution layers implemented using Groupy (Cohen & Welling, 2016).

We extracted the last convolutional activation of this pretrained equivariant VGG19 group after rescaling the input images by $2^s$, where $s = -3, -2.5, ..., 1.5$. For efficiency, the scales that increased the image size to greater than $1,024^2$ pixels, were discarded. Subsequently, the nonlinear embedding method proposed in Vedaldi & Zisserman (2012) was applied such that the feature dimensions were three times larger. As this embedding is a point-wise function, the output can be regarded as three times the direct sum of the original representations when considering the group action.

The local feature dimension was then reduced using PCA for the existing method and the proposed invariant PCA for the proposed method. We applied BP and iBP with dimensions of 1,024, VLAD with dimensions of 512 and 1,024 components, FV with 512 dimensions and 512 components, and VLAT with 256 dimensions and 8 components. We applied the proposed BP and iBP with the same settings, and VLAD and VLAT with eight times the number of components.

Training code given pretrained VGG19 was implemented using MATLAB. The linear SVM implemented in LIBLINEAR (Fan et al., 2008) was used to evaluate the average test accuracy. Furthermore, to validate

Table 6: Comparison of accuracy using existing feature coding methods with augmented training data. Scores in the parentheses correspond to those of the proposed invariant features cited from Table 3.

| Methods | Dim. | FMD | UIUC |
|---------|------|-----|------|
| BP | 525k | $83.00 \pm 1.51$ ($83.34\pm1.48$) | $81.11 \pm 1.46$ ($81.48\pm1.75$) |
| iBP | 525k | $83.10 \pm 1.38$ ($83.46\pm1.27$) | $82.87 \pm 2.01$ ($83.33\pm1.80$) |
| VLAD | 525k | $82.72 \pm 1.66$ ($82.42\pm1.74$) | $80.56 \pm 1.75$ ($81.94\pm1.40$) |
| VLAT | 525k | $82.66 \pm 1.27$ ($81.88\pm1.34$) | $81.67 \pm 1.89$ ($82.13\pm1.64$) |
| FV | 525k | $81.44 \pm 1.60$ | $80.74 \pm 2.46$ |

Table 7: Irreducible representations of the D6 group.

| Rep. | $r$ | $m$ |
|------|-----|-----|
| $\tau_{1,1}$ | 1 | 1 |
| $\tau_{1,-1}$ | 1 | -1 |
| $\tau_{-1,1}$ | -1 | 1 |
| $\tau_{-1,-1}$ | -1 | -1 |
| $\tau_{2a}$ | $\begin{bmatrix} \cos(\pi/3) & -\sin(\pi/3) \\ \sin(\pi/3) & \cos(\pi/3) \end{bmatrix}$ | $\begin{bmatrix} 0 & -1 \\ 1 & 0 \end{bmatrix}$ |
| $\tau_{2b}$ | $\begin{bmatrix} \cos(2\pi/3) & -\sin(2\pi/3) \\ \sin(2\pi/3) & \cos(2\pi/3) \end{bmatrix}$ | $\begin{bmatrix} 0 & -1 \\ 1 & 0 \end{bmatrix}$ |

that the proposed feature is D4 invariant, we used the same training data and evaluated the accuracy by augmenting the test data eight-fold using the D4 group.

Tables 3 and 4 show the accuracy for the test as well as the augmented test datasets. Our method demonstrated better accuracy than non-invariant methods with the dimensions of the invariant BP and iBP approximately 1/7 of the original dimensions. Thus, we obtained features with significantly smaller dimensions and higher performances. Note that the higher accuracy is not a simple effect of dimensionality reduction because in general, the accuracy decreases as the dimension is reduced. We observed that iBP with 400 local feature dimensions (80k global features) scored 81.76 on FMD and 81.30 on UIUC. Furthermore, compact bilinear pooling with 82k feature dimensions scored 81.54 on FMD and 79.81 on UIUC. Therefore, the result is significant as Inv iBP shows better accuracy **despite** the lower feature dimension.

This table also shows that the existing methods exhibit poor performance on the augmented test data, but the proposed methods demonstrate performance with scores similar to the original. These results suggest that the existing methods use information that is not related to image content, reflecting dataset bias instead. Our method can discard this bias and focus on the image content.

As an ablation study, we conducted experiments in which PCA and k-means were trained using the proposed invariant feature modeling and existing feature coding methods were then applied.

Table 5 demonstrates that the proposed invariant feature modeling, combined with the standard feature coding method, does not demonstrate better accuracy than the original methods. This result shows that only performance is enhanced when the proposed invariant feature modeling is combined with invariant feature coding.

We also compared our results with those of data augmentation by training the existing non-invariant feature coding with eight-fold data augmentation. We conducted this by applying all transformations to the input images and made the number of training samples $|G|$ times as large. Considering the much higher training cost, experiments were conducted on the FMD and UIUC datasets, which have a smaller number of training images.

Table 6 shows that the accuracy is comparable or a little lower than the accuracy of the proposed invariant feature coding methods. This result is consistent with the argument of Theorem 1 that invariant features learned with the original training data are equivalent to non-invariant features learned with the augmented

Table 8: Irreducible representations of the tensor products of irreducible representations of the D6 group.

| | $\tau_{1,1}$ | $\tau_{1,-1}$ | $\tau_{-1,1}$ | $\tau_{-1,-1}$ | $\tau_{2a}$ | $\tau_{2b}$ |
|---|---|---|---|---|---|---|
| $\tau_{1,1}$ | $\tau_{1,1}$ | $\tau_{1,-1}$ | $\tau_{-1,1}$ | $\tau_{-1,-1}$ | $\tau_{2a}$ | $\tau_{2b}$ |
| $\tau_{1,-1}$ | $\tau_{1,-1}$ | $\tau_{1,1}$ | $\tau_{-1,-1}$ | $\tau_{-1,1}$ | $\tau_{2a}$ | $\tau_{2b}$ |
| $\tau_{-1,1}$ | $\tau_{-1,-1}$ | $\tau_{-1,1}$ | $\tau_{1,-1}$ | $\tau_{1,1}$ | $\tau_{2b}$ | $\tau_{2a}$ |
| $\tau_{-1,-1}$ | $\tau_{-1,-1}$ | $\tau_{-1,1}$ | $\tau_{1,-1}$ | $\tau_{1,1}$ | $\tau_{2b}$ | $\tau_{2a}$ |
| $\tau_{2a}$ | $\tau_{2a}$ | $\tau_{2a}$ | $\tau_{2b}$ | $\tau_{2b}$ | $\tau_{1,1} \oplus \tau_{1,-1} \oplus \tau_{2b}$ | $\tau_{-1,1} \oplus \tau_{-1,-1} \oplus \tau_{2a}$ |
| $\tau_{2b}$ | $\tau_{2b}$ | $\tau_{2b}$ | $\tau_{2a}$ | $\tau_{2a}$ | $\tau_{-1,1} \oplus \tau_{-1,-1} \oplus \tau_{2a}$ | $\tau_{1,1} \oplus \tau_{1,-1} \oplus \tau_{2b}$ |

Table 9: Comparison of test accuracy using D6-equivariant CNN.

| Methods | Dim. | FMD | DTD | UIUC | CUB | Cars |
|---|---|---|---|---|---|---|
| BP | 525k | 77.26±1.69 | 73.46±0.87 | 75.93±2.58 | 72.66 | 80.15 |
| iBP | 525k | 77.18±1.42 | 73.51±0.65 | 75.56±1.70 | 71.19 | 81.13 |
| VLAD | 525k | 75.80±1.55 | 71.65±0.99 | 70.19±3.23 | 64.69 | 78.44 |
| VLAT | 525k | 75.70±1.40 | 72.60±0.85 | 72.31±2.07 | 69.92 | 78.59 |
| FV | 525k | 75.24±1.42 | 72.37±0.92 | 74.91±3.48 | 63.06 | 75.55 |
| Inv BP (ours) | 55k | 80.26±1.70 | 75.41±1.01 | 79.26±2.93 | 78.77 | 80.92 |
| Inv iBP (ours) | 55k | **80.36±1.48** | **75.86±0.55** | **80.09±2.66** | **79.74** | **83.60** |
| Inv VLAD (ours) | 525k | 78.00±1.17 | 72.76±0.71 | 73.98±2.52 | 76.12 | 82.47 |
| Inv VLAT (ours) | 525k | 78.06±1.45 | 74.15±0.77 | 75.65±2.73 | 77.59 | 80.15 |

training data. Therefore, our method exploits the advantages of data augmentation with lower feature dimensions and lower training costs.

As for computation time, using an Intel Xeon E5-2698v4 x2 20 Core, 2.2 GHz CPU it takes 13 seconds to extract the training features and 61 seconds to learn SVM to train BP on UIUC, 9.4 seconds to extract the training features and 2.1 seconds to learn SVM to train Inv BP on UIUC in Table 3, and 81 seconds to extract the training features and 130 seconds to learn SVM to train BP on the augmented UIUC in Table 6. The reduced number of feature dimensions by the proposed method contributes to the training efficiency.

### 5.1.2 Results on D6 Group

Furthermore, we applied our method to the D6 group, which was more complex than the D4 group. The D6 group consists of $\pi/3$ rotations $r$ and an image flipping $m$ with $|G| = 12$. The irreducible representations and decomposition of the tensor products for this group are summarized in Tables 7 and 8, respectively.

To obtain the local feature in which D6 acts orthogonally, CNN was pretrained with HexaConv (Hoogeboom et al., 2018). HexaConv models the input images in the hexagonal axis and applies D6 group-equivariant convolutional layers to construct the CNN. Because D6 acts as a permutation of the positions along the hexagonal axis, the convolutional activations are D6-equivariant. Therefore, we used this convolutional activation as the input for our coding methods.

The group equivariant CNNs with the VGG19 architecture were pretrained with convolutional filter sizes of $18, 37, 74, 148, 148$ instead of $64, 128, 256, 512, 512$ as the local feature extractor. Furthermore, batch normalization layers were added to each convolution layer to accelerate the training speed. Therefore, the dimensionality of the local features is $148 \times 12$. The model was trained using the ILSVRC2012 dataset (Deng et al., 2009). Because the input image size increases when the axes are changed from Euclidean to hexagonal, a $160 \times 160$ image was randomly cropped from the original image rescaled to $192 \times 192$ for training. The remaining settings followed those of the original VGG.

We extracted the last convolutional activation of this pretrained model after rescaling the input images by $2^s$, where $s = -3, -2.5, ..., 1.5$. For efficiency, the scalings that increased the image size to greater than $512^2$ pixels were discarded. Subsequently, the nonlinear embedding method proposed in Vedaldi & Zisserman (2012) was applied. The dimensions and number of components used for the coding methods followed

Table 10: Comparison of accuracy on Resnet50. The score with '*' denotes the score reported in Li et al. (2018).

| Method | iSQRT-COV* (Resnet50) | iSQRT-COV (equivariant Resnet50) | Inv iSQRT-COV (equivariant Resnet50) (ours) |
|---|---|---|---|
| Dim. | 32k | 265k | 25k |
| Top-1 Err. | 22.14 | 21.65 | **21.02** |
| Top-5 Err. | 6.22 | 5.92 | **5.47** |

Table 11: Comparison of accuracy on Resnet101. The score with '*' denotes the score reported in Li et al. (2018).

| Method | iSQRT-COV* (Resnet101) | iSQRT-COV (equivariant Resnet101) | Inv iSQRT-COV (equivariant Resnet101) (ours) |
|---|---|---|---|
| Dim. | 32k | 265k | 25k |
| Top-1 Err. | 21.21 | 20.45 | **19.98** |
| Top-5 Err. | 5.68 | 5.35 | **4.96** |

those used for the D4 experiments. The pretraining was implemented by Pytorch and hexaconv library (Hoogeboom et al., 2018), and feature coding and classifier training was implemented with MATLAB.

In Table 9, the proposed coding methods consistently demonstrate better performance than non-invariant methods although the overall accuracy is lower than the D4 case, which may arise from the discriminative performance of the local feature extractor. Furthermore, the dimensionality of invariant BP and iBP is smaller than the dimensionality for D4. This is because the D6 group represents more complex transformations than the D4 group; thus, the number of invariants with respect to the D6 group is smaller than the number with respect to the D4 group.

Therefore, the proposed framework was also effective for the D6 group.

## 5.2 Experiment with end-to-end model

The proposed invariant BP was then applied to an end-to-end learning framework.

### 5.2.1 Results on D4 Group with iSQRT-COV

We constructed a model based on iSQRT-COV (Li et al., 2018), which is a variant of BP that demonstrates good performance and stable training. Group equivariant CNNs with Resnet50 and 101 (He et al., 2016) architecture were used for the local feature extractor, where the filter sizes were reduced, as in the case of VGG19. The iSQRT-COV and the proposed invariant iSQRT-COV were used instead of global average pooling. We compared iSQRT-COV with Resnet50, iSQRT-COV with D4-equivariant Resnet50, and the proposed invariant iSQRT-COV with D4-equivariat Resnet50.

All the models were learned, including the feature extractor, using a momentum grad with an initial learning rate of 0.1, momentum of 0.9, and weight decay rate of 1e-4 for 65 epochs with a batch size of 160. The learning rate was multiplied by 0.1 at 30, 45, and 60 epochs. The top-1 and top-5 test errors were then evaluated, using the average score for the original and flipped images for the evaluation. The training code was implemented with Pytorch, Groupy and fast-MPN-COV libraries (Li et al., 2018).

Tables 10 and 11 show that although iSQRT-COV achieves accuracy using the local features extracted from equivariant CNN, our Inv iSQRT-COV demonstrates better accuracy with smaller feature dimension.

The above pre-trained models were further fine-tuned and the accuracy was evaluated on fine-grained recognition datasets: CUB, Cars, and Aircraft (Maji et al., 2013). The Aircraft dataset consists of 100 categories with 6,667 training images and 3,333 test images. Following the settings of existing studies, each input image was resized to $448 \times 448$ for fine-tuning and evaluation for CUB and Cars, and the other images were resized

Table 12: Comparison of accuracy using fine-tuning on Resnet50. The score with '*' denotes the score reported in Li et al. (2018).

| Method | iSQRT-COV* (Resnet50) | iSQRT-COV (equivariant Resnet50) | Inv iSQRT-COV (equivariant Resnet50) (ours) |
|---|---|---|---|
| Dim. | 32k | 265k | 25k |
| CUB | **88.1** | 87.8±0.19 | 87.9±0.26 |
| Cars | 92.8 | **93.8±0.11** | 93.4±0.16 |
| Aircraft | 90.0 | 91.4±0.28 | **91.7±0.32** |

Table 13: Comparison of accuracy using fine-tuning on Resnet101. The score with '*' denotes the score reported in Li et al. (2018).

| Method | iSQRT-COV* (Resnet101) | iSQRT-COV (equivariant Resnet101) | Inv iSQRT-COV (equivariant Resnet101) (ours) |
|---|---|---|---|
| Dim. | 32k | 265k | 25k |
| CUB | **88.7** | 44.7±2.3 | 88.5±0.25 |
| Cars | 93.3 | 56.0±7.9 | **93.9±0.18** |
| Aircraft | 91.4 | 69.1±2.0 | **92.1±0.26** |

to $512 \times 512$, further cropping the $448 \times 448$ center image for the Aircraft dataset. The training setting followed the original setting of the iSQRT-COV (Li et al., 2018). For each setting, we evaluated the models ten times and evaluated the average test accuracy. Tables 12 and 13 show the results. As for ResNet50, the proposed Inv iSQRT-COV displays comparable or better accuracy than the reported iSQRT-COV and similar accuracy to iSQRT-COV with equivariant ResNet50 with much smaller feature dimension. Furthermore, iSQRT-COV with equivariant ResNet101 overfits the training data and displays much lower accuracy than the proposed Inv iSQRT-COV ResNet101. This result suggests that our invariant feature coding contributes to the stability of gradient-based training and results in the exploitation of complex local feature extractors with high recognition accuracy.

As for the computation time, when using 8 A100 GPUs it takes 0.17 seconds/batch to train iSQRT-COV (Resnet50), 0.72 seconds/batch to train iSQRT-COV (equivariant Resnet50) and 0.45 seconds/batch to train the proposed Inv iSQRT-COV (equivariant Resnet50) on ImageNet. Though equivariant Resnet50 models take more time than the original Resnet50, which mainly results from feature extractor module instead of the feature coding module we target, we can reduce the computation time by the proposed invariant coding method compared to iSQRT-COV (equivariant Resnet50). The reduced complexity arises from (i) reduction of the feature dimension that we calculate the bilinear feature, and (ii) reduction of the size of the final fully-connected layer.

Subsequently, to visualize attention, GradCAM++ (Chattopadhay et al., 2018) was used after the dimension-reduction layer (Figure 5). In the existing methods (iSQRT-COV with Resnet50 and equivariant Resnet50), the shape of the heatmap changes when rotation and horizontal flip are applied to the input image. The shape of the heatmap is relatively preserved with the proposed invariant coding method. This result indicates that the baseline methods learned in a standard manner do not learn the invariance correctly, whereas our method was trained to both demonstrate high accuracy and preserve invariance, which is consistent with the result for the case of fixed local features. Furthermore, the heatmaps are quantatively compared using maximum mean discrepancy (MMD) by (i) inverting flipped and rotated heatmaps to match the original heatmaps, (ii) applying l1-normalization to the heatmaps, and (iii) calculating the MMD between the inverted heatmaps and heatmaps for the original image with the Gaussian kernel function $\exp(-\frac{\|r_1 - r_2\|^2}{\sigma^2})$, where $r_i$s are two-dimensional positions of the image pixels normalized within $[0, 1]$, and $\sigma$ is set to 0.1. We can see that the proposed method shows a lower MMD value, which means that the shape of the heatmaps are more similar to each other. Plots of the results on the other images are available in the Supplementary Materials section.

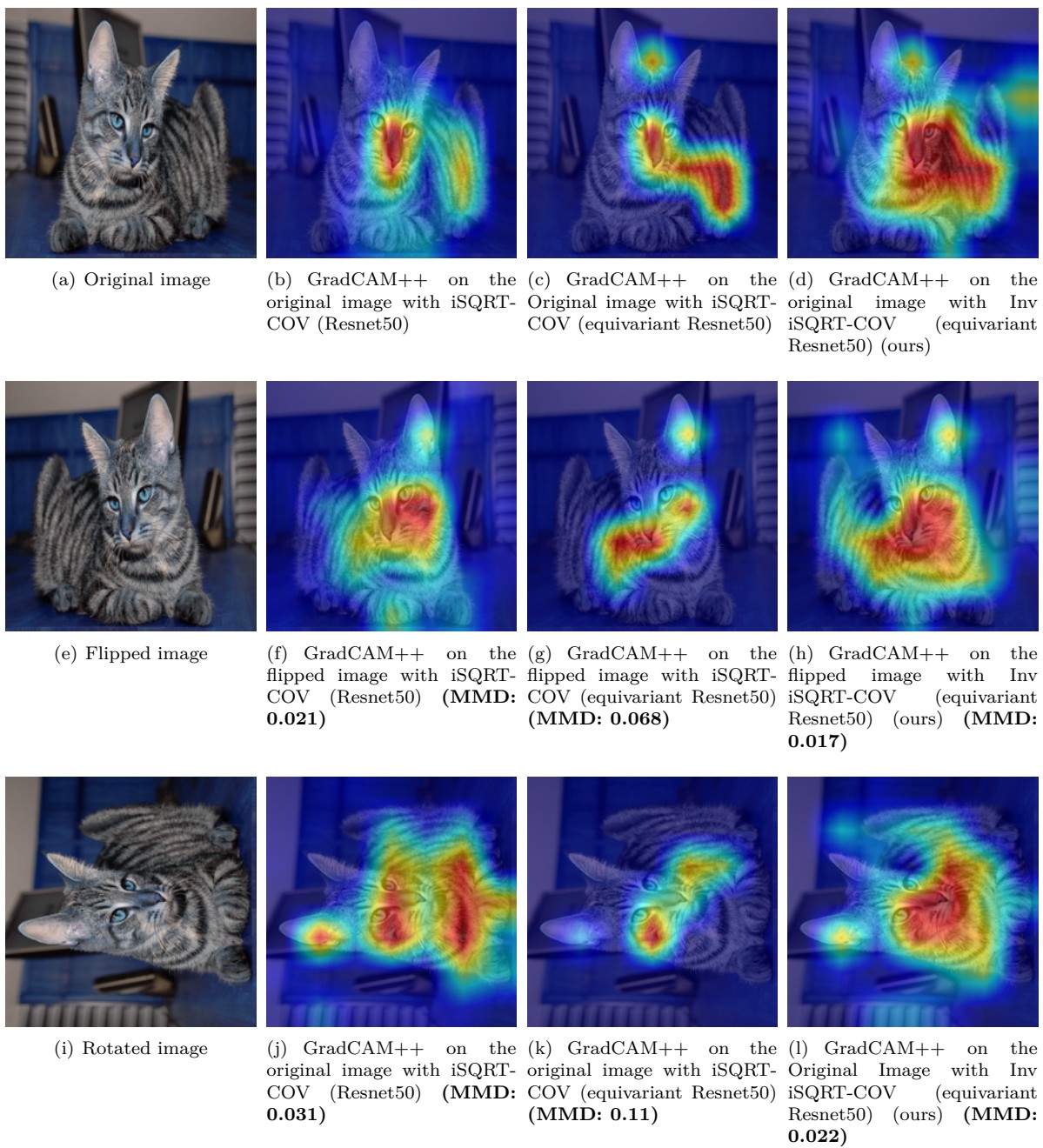

Figure 5: Comparison of GradCAM++ on the transformed input image. MMD indicates maximum mean discrepancy to the heatmaps of the original image.

### 5.2.2 Results on the other groups

The proposed method was then applied to different groups.

First, we used D6 group with HexaConv (Hoogeboom et al., 2018) as in Section 5.1.2 with D6-equivariant Resnet18 as the base feature extractor. Resnet18 was used to reduce memory consumption. We then compared the standard Resnet18, D6-equivariant Resnet18, iSQRT-COV with D6-equivariant Resnet18, and the proposed invariant iSQRT-COV with D6-equivariant Resnet18.

Table 14: Comparison of accuracy for the end-to-end D6-equivariant model.

| Method | Resnet18 | equivariant Resnet18 | iSQRT-COV (equivariant Resnet18) | Inv iSQRT-COV (equivariant Resnet18) (ours) |
|---|---|---|---|---|
| Dim. | 0.5k | 1.8k | 395k | 22k |
| Top-1 Err. | 30.24 | 28.93 | 27.27 | **26.42** |
| Top-5 Err. | 10.92 | 9.99 | 9.46 | **8.45** |

Table 15: Comparison of accuracy for the end-to-end C8-equivariant model.

| Method | Resnet18 | equivariant Resnet18 | iSQRT-COV (equivariant Resnet18) | Inv iSQRT-COV (equivariant Resnet18) (ours) |
|---|---|---|---|---|
| Dim. | 0.5k | 1.5k | 131k | 10k |
| Top-1 Err. | 30.24 | 27.32 | 25.33 | **25.25** |
| Top-5 Err. | 10.92 | 8.76 | 7.90 | **7.86** |

To learn the iSQRT-COV models, a momentum grad with an initial learning rate of 0.1, momentum of 0.9, and weight decay rate of 1e-4 were used for 65 epochs with a batch size of 160. The learning rate was further multiplied by 0.1 at 30, 45, and 60 epochs. Following the original setting, we learned the Resnet models using a momentum grad with an initial learning rate of 0.1, momentum of 0.9, and weight decay rate of 1e-4 for 100 epochs with a batch size of 256. The learning rate was multiplied by 0.1 at 30, 70, and 90 epochs, and then the top-1 and top-5 test errors were evaluated. The training code was implemented with Pytorch, hexaconv and fast-MPN-COV libraries.

Table 14 demonstrates that the proposed invariant features show good accuracy with low feature dimension.

When considering the discrete rotation group, the proposed invariant bilinear pooling was reduced to the squared norm of the Fourier coefficients. The Fourier coefficients were then applied to the C8 group that consists of $\pi/8$ rotations. The implementation provided by Weiler & Cesa (2019) was used to implement the C8-equivariant networks. The standard Resnet18, C8-equivariant Resnet18, C8-equivariant Resnet18 with iSQRT-COV pooling, and the proposed C8-equivariatn Resnet18 with invariant iSQRT-COV pooling were then compared. The training setting followed that of the D6 group.

Table 15 demonstrates a similar tendency for the D4 and D6 cases. Therefore, the proposed method is effective for several invariance settings.

### 5.2.3 Results on the other coding method

In this section, we further apply our method to another variant of bilinear pooling.

Scaling eigen branch (SEB) (Song et al., 2023) is a method that multiplies improved bilinear pooling with $1 + \sqrt{\sum_{i=1}^{d_{\text{local}}} \lambda_i e^{-2\lambda_i}}$ to enhance the importance of features with smaller variance, where $\lambda_i$ denotes the $i$-th eigenvalue of the bilinear matrix. Although the original method calculates the features using singular value decomposition, to reduce the computation cost and increase the stability, we calculated the features using $\left(1 + \sqrt{\text{trace}\left(\Sigma e^{-2\Sigma}\right)}\right)$ iSQRT$(\Sigma)$, where $\Sigma$ is the input covariance feature. The computation does not require singular value decomposition. We refer to this method as iSQRT-SEB and compare iSQRT-SEB Resnet50, D4-equivariant Resnet50 with iSQRT-SEB (denoted by "iSQRT-SEB") and the proposed D4-equivariant Resnet50 with invariant iSQRT-SEB (denoted by "Inv iSQRT-SEB (ours)"). For iSQRT-SEB with D4-equivariant Resnet50, the global feature dimension was reduced to 32k by applying invariant PCA, considering memory limitations. The training strategy was the same as that of iSQRT-COV in the D4 group. The DifferentialSVD library (Song et al., 2023) was used for implementing SEB. Tables 16 and 17 display the results. Although the overall accuracy is lower than that of iSQRT, showing the same tendency as in the original paper, the tendency is the same as that of the proposed method which has better accuracy.

Therefore, invariant feature coding is effective for end-to-end training.

Table 16: Comparison of accuracy for iSQRT-SEB on Resnet50.

| Method | iSQRT-SEB (Resnet50) | iSQRT-SEB (equivariant Resnet50) | Inv iSQRT-SEB (equivariant Resnet50) (ours) |
|---|---|---|---|
| Dim. | 32k | 32k | 25k |
| Top-1 Err. | 25.20 | 24.62 | **23.77** |
| Top-5 Err. | 7.92 | 7.71 | **7.14** |

Table 17: Comparison of accuracy for iSQRT-SEB on Resnet101.

| Method | iSQRT-SEB (Resnet101) | iSQRT-SEB (equivariant Resnet101) | Inv iSQRT-SEB (equivariant Resnet101) (ours) |
|---|---|---|---|
| Dim. | 32k | 32k | 25k |
| Top-1 Err. | 23.84 | 23.87 | **22.96** |
| Top-5 Err. | 7.7 | 7.11 | **6.62** |

## 6 Conclusion

In this study, we proposed a feature-coding method that considers image information preserving transformations. Based on group representation theory, we propose a guideline used to first construct a feature space in which groups act orthogonally to calculate the invariant vector. Subsequently, a novel model learning method and coding methods are introduced to explicitly consider group actions. We applied our method to image classification datasets and demonstrated that the proposed model can yield high accuracy while preserving invariance. This study provides a novel framework for constructing an invariant feature.

**Acknowledgements**

This work was partially supported by JSPS KAKENHI Grant Number JP19176033 and JP19K20290, JST Moonshot R&D Grant Number JPMJPS2011, CREST Grant Number JPMJCR1403 and JPMJCR2015 and Basic Research Grant (Super AI) of Institute for AI and Beyond of the University of Tokyo. We would like to thank Atsushi Kanehira, Takuhiro Kaneko, Toshihiko Matsuura, Takuya Nanri and Masatoshi Hidaka for the helpful discussion.

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

## A  Another Proof of Theorem 1

In this section, the proof of Theorem 1is provided.

*Proof.* Denoting the loss function as $L(W)$, the subgradient $\partial L$ at $W_0$ is defined as the set,

$$\partial L(W_0) = \{c | f(W) \geqq f(W_0) + \langle c, W - W_0 \rangle \text{ for } \forall W\}. \tag{8}$$

From this definition, $W_0$ minimizes $L$ if and only if $0 \in \partial L(W_0)$. From the assumption in Theorem 1, $L(W) = L(\pi(g)W)$ for any $W, g$. Therefore, $\partial L(\pi(g)W_0) = \pi(g)^\top \partial L(W_0)$ and $0 \in \partial L(W_0) \iff 0 \in \partial L(\pi(g)W_0)$. From the definition of subgradient, when $0 \in \partial L(W_0)$, it follows that

$$\partial L \left( \frac{1}{|G|} \sum_{g \in G} \pi(g)W_0 \right) \supset \bigcap_g \partial L(\pi(g)W_0) \ni 0. \tag{9}$$

The inclusion comes from the fact that if $c \in \bigcap_g \partial L(\pi(g)W_0)$, then $L(W) \geqq \frac{1}{|G|} \sum_{g \in G} L(\pi(g)W_0) + \langle c, W - \frac{1}{|G|} \sum_{g \in G} \pi(g)W_0 \rangle \geqq L(\frac{1}{|G|} \sum_{g \in G} \pi(g)W_0) + \langle c, W - \frac{1}{|G|} \sum_{g \in G} \pi(g)W_0 \rangle$ Therefore, $\frac{1}{|G|} \sum_{g \in G} \pi(g)W_0$ also minimizes $L$, which is invariant under $\pi(g)$. $\qquad\square$

## B  Extension of Theorem 1

**Non-orthogonal case**   When we omit the restriction on the orthogonality of $\pi$,

$$\frac{1}{M|G|} \sum_{m=1}^M \sum_{g \in G} l \left( \text{Re} \left( \sum_{t=1}^T \left( W^{(t)} \right)^\top \tau_t(g^{-1}) v_m^{(t)} \right), y_m \right)$$

$$= \frac{1}{M|G|} \sum_{m=1}^M \sum_{g \in G} l \left( \text{Re} \left( \sum_{t=1}^T \left( \tau_t(g) W^{(t)} \right)^\top v_m^{(t)} \right), y_m \right), \tag{10}$$

$$\tag{11}$$

does not follow. However, matrix $T$ can be calculated such that $T\pi(g)T^{-1}$ becomes orthogonal for all $g$. Therefore, if we apply an appropriate coordinate transformation beforehand, we can apply Theorem 1.

**Infinite case** The finite group can be extended to a compact group. The compact group is parameterized by a closed bounded set, and $\circ$ and $-1$ are continuous with respect to these parameters. Many results regarding the representation of a finite group also hold for a compact group when $\frac{1}{|G|}\sum_{g\in G}$ is replaced with $\int dg$, where $dg$ is a special measure called Haar measure. With this measure, the following holds.

$$P_{\tau_t} = \dim(\tau_t)\int dg\overline{\chi_{\tau_t}(g)}\pi(g), \tag{12}$$

and

$$P_{\mathbf{1}} = \int dg\pi(g). \tag{13}$$

Theorem 1 can be rewritten as

**Theorem 2.** *When we assume that the compact group $\mathcal{G}$ acts as an orthogonal representation $\pi$ on $\mathbb{R}^d$ and preserves the distribution of the training data $P$, the solution to the L2-regularized convex loss minimization,*

$$\arg\min_{W\in\mathbb{R}^{d\times|C|}} \frac{\lambda}{2}\|W\|_F^2 + E_{(x,y)\sim P}[l(W^\top x, y)] \tag{14}$$

*is $\mathcal{G}$-invariant, implying that $\pi(g)W = W$ for any $g$ and $P_{\mathbf{1}}W = W$.*

## C  Invariant PCA when the irreducible representations are not real.

When some $\tau$ has complex elements, Algorithm 1 cannot be applied directly because the intertwining operator and covariance matrices become complex. In this case, we couple $\tau_t$ with $\overline{\tau_t}$ into $\tau' \simeq \tau_t \oplus \overline{\tau_t}$. Because $\chi_{\overline{\tau_t}} = \overline{\chi_{\tau_t}}$, the multiplicities of $\tau_t$ and $\overline{\tau_t}$ in the decomposition are equal because of Eq.(6), and the projected vectors are complex conjugates of each other because of Eq.(7). Thus, $\sqrt{2}$ times the concatenation of the real and imaginary parts constitute the $\tau'$-th element. In Algorithm 1, $\tau_t$ is replaced with $\tau_t'$ to obtain an invariant PCA for the general case.

## D  Proof that the dimension of Invariant VLAD and VLAT is the same as that of original feature

Using $C$ components, $1_\mu$ can be decomposed into $C\tau_{1,1} \oplus C\tau_{1,-1} \oplus C\tau_{-1,1} \oplus C\tau_{-1,-1} \oplus 2C\tau_2$. When the first- or second-order statistics are decomposed as $n_{1,1}\tau_{1,1} \oplus n_{1,-1}\tau_{1,-1} \oplus n_{-1,1}\tau_{-1,1} \oplus n_{-1,-1}\tau_{-1,-1} \oplus n_2\tau_2$, the multiplicity of $\tau_{1,1}$ of $(C\tau_{1,1} \oplus C\tau_{1,-1} \oplus C\tau_{-1,1} \oplus C\tau_{-1,-1} \oplus 2C\tau_2) \otimes (n_{1,1}\tau_{1,1} \oplus n_{1,-1}\tau_{1,-1} \oplus n_{-1,1}\tau_{-1,1} \oplus n_{-1,-1}\tau_{-1,-1} \oplus n_2\tau_2)$ is $C(n_{1,1} + n_{1,-1} + n_{-1,1} + n_{-1,-1} + 2n_2)$, which is the same as the dimension of the original feature.

## E  GradCAM++ on the other images

GradCAM++ results on the other images are plotted in Figures 6 and 7.

## F  Results on SIFT features

In this section, we report the results using the SIFT feature in which D4 acts orthogonally. The irreducible decomposition of the SIFT feature is described in Section F.1. The accuracy of the image recognition datasets was evaluated in Section Section F.2.

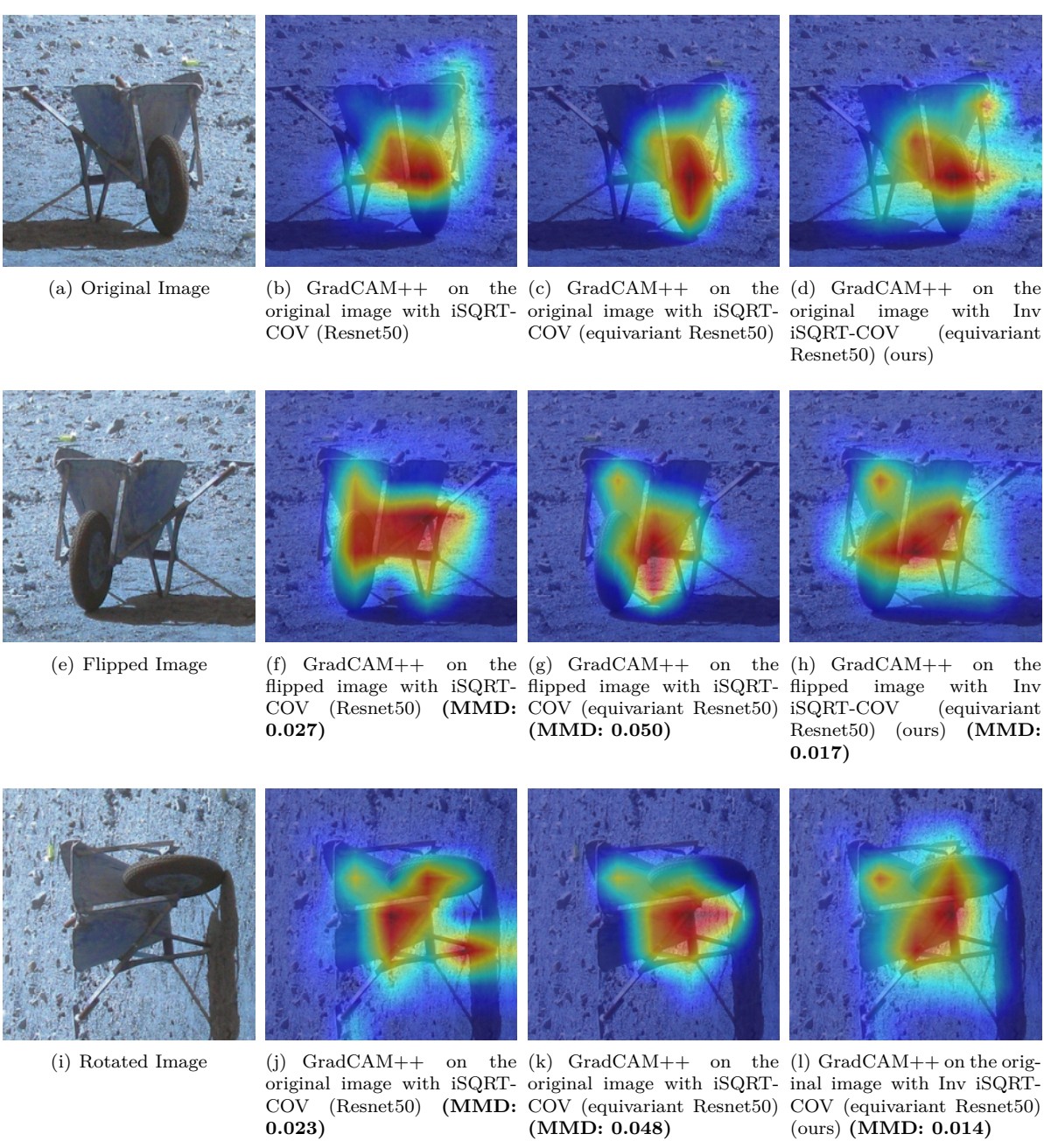

Figure 6: Comparison of GradCAM++ on the transformed input image. MMD indicates maximum mean discrepancy to the heatmaps of the original image.

## F.1 Irreducible decomposition of SIFT

An overview of the SIFT feature is plotted in Figure 8, where a D4 group acts as a permutator on both $\{a, b, c, d, e, f, g, h\}$ and $1...16$. We can further decompose these into permutations on $\{a, c, e, g\}$: $\{b, d, f, h\}$, $\{1, 4, 13, 16\}$, $\{6, 7, 10, 11\}$, and $\{2, 3, 5, 8, 9, 12, 13, 14\}$. These permutations can be decomposed as: $\tau_{1,1} \oplus \tau_{-1,1} \oplus \tau_2$, $\tau_{1,1} \oplus \tau_{-1,-1} \oplus \tau_2$, $\tau_{1,1} \oplus \tau_{-1,-1} \oplus \tau_2$, $\tau_{1,1} \oplus \tau_{-1,-1} \oplus \tau_2$, and $\tau_{1,1} \oplus \tau_{-1,1} \oplus \tau_{1,-1} \oplus \tau_{-1,-1} \oplus \tau_2$ respectively, which can be calculated using the characteristic functions. Thus, the permutations on SIFT can be decomposed into $(2\tau_{1,1} \oplus \tau_{-1,1} \oplus \tau_{-1,-1} \oplus 2\tau_2) \otimes (3\tau_{1,1} \oplus \tau_{1,-1} \oplus \tau_{-1,1} \oplus 3\tau_{-1,-1} \oplus 4\tau_2) = 18\tau_{1,1} \oplus 14\tau_{1,-1} \oplus 14\tau_{-1,1} \oplus 18\tau_{-1,-1} \oplus 32\tau_2$.

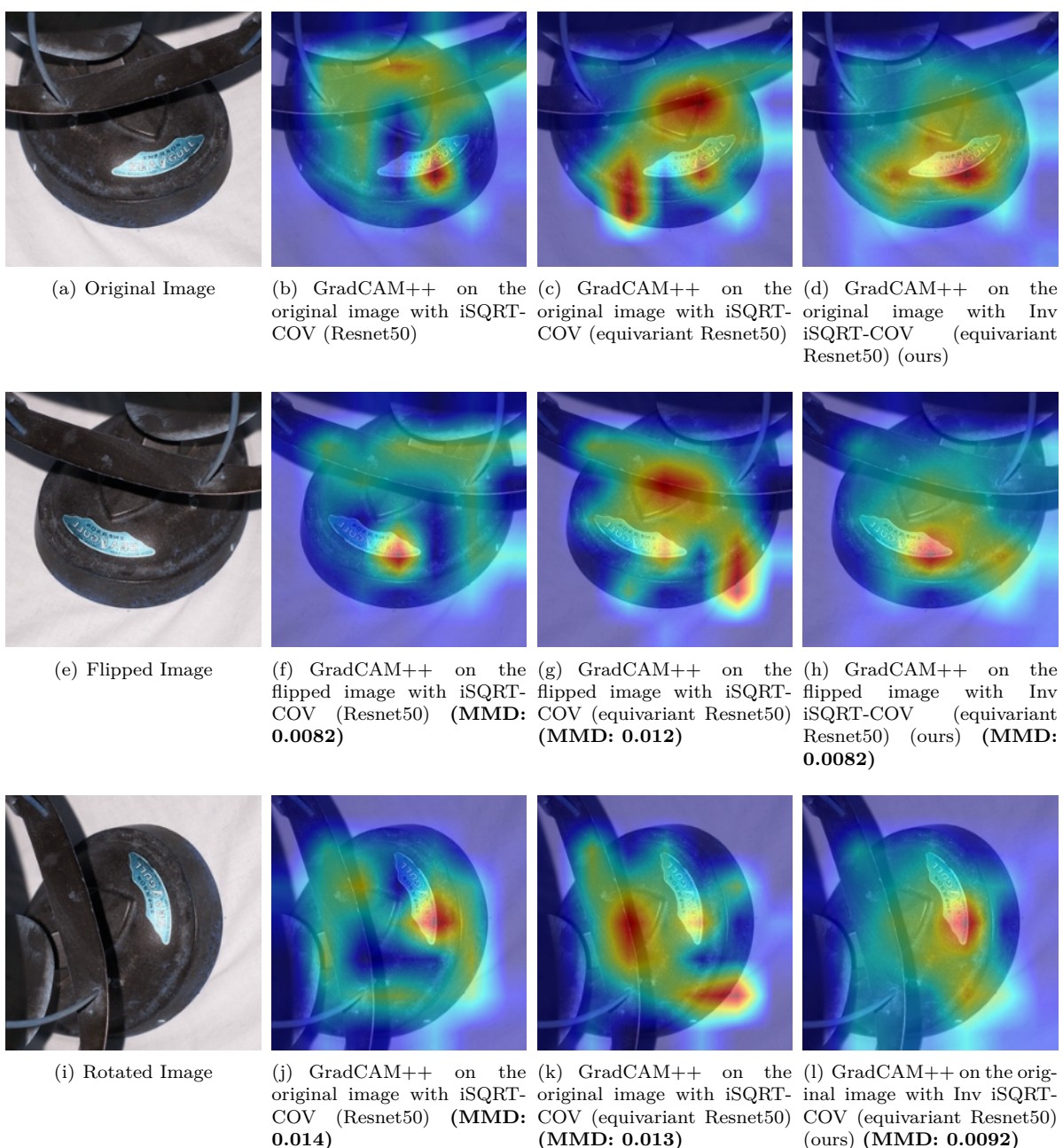

Figure 7: Comparison of GradCAM++ on the transformed input image. MMD indicates maximum mean discrepancy to the heatmaps of the original image.

## F.2 Experimental Results

The methods were evaluated on (FMD) (Sharan et al., 2013), (DTD) (Cimpoi et al., 2014), (UIUC) (Liao et al., 2013), and CUB (Welinder et al., 2010)). The evaluation protocol was the same as that for the VGG-feature.

The dense SIFT feature was extracted from multi-scale images, as in the case of VGG, and then nonlinear homogeneous mapping (Vedaldi & Zisserman, 2012) was applied to make the feature dimension three times larger.

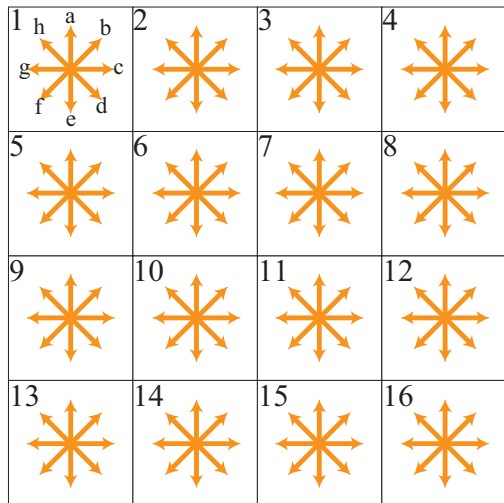

Figure 8: Overview of the SIFT feature.

Table 18: Comparison of accuracy using SIFT features.

| Methods | Dim. | Test Accuracy | | | Augmented Test Accuracy | | |
|---|---|---|---|---|---|---|---|
| | | FMD | DTD | UIUC | FMD | DTD | UIUC |
| BP | 33k | 49.84±1.57 | 51.54±0.93 | 56.02±2.40 | 43.36±1.19 | 41.55±0.73 | 42.50±2.23 |
| VLAD | 262k | 60.11±1.41 | 59.37±1.23 | 58.70±3.50 | 54.68±1.17 | 51.54±0.90 | 46.11±2.01 |
| VLAT | 525k | 58.09±1.40 | 58.74±0.84 | 59.72±2.87 | 52.01±1.25 | 50.40±0.81 | 47.85±1.65 |
| FV | 262k | 61.84±1.87 | 61.20±0.96 | 64.26±3.30 | 56.78±0.78 | 53.77±0.78 | 51.89±2.29 |
| Inv BP (ours) | 8k | 55.19±1.62 | 54.78±1.37 | 60.74±2.84 | 55.18±1.75 | 54.79±1.36 | 60.74±3.02 |
| Inv VLAD (ours) | 262k | **67.53±1.03** | 64.50±1.01 | **68.98±2.55** | **67.55±1.11** | 64.53±1.03 | **69.04±2.58** |
| Inv VLAT (ours) | 525k | 66.79±1.47 | **64.60±1.12** | 67.50±4.15 | 66.75±1.48 | **64.61±1.07** | 67.57±4.10 |

The dimension was reduced to 256 prior to applying BP, VLAD with 1,024 components, FV with 512 components, and VLAT with 16 components. We also applied the proposed invariant BP with the same setting, and VLAD and VLAT with eight times the number of components.

Table 18 shows a tendency similar to the results of the VGG-feature. Our methods demonstrate better performance than existing methods for both the original test data and augmented test data.

