# OpenReview forum: "Invariant Feature Coding using Tensor Product Representation"
_TMLR — Accepted by TMLR_

### Review · Reviewer_xXoe · 2023-03-21

**Summary Of Contributions:**

The paper considers learning invariant global features in convolutional neural networks. Here the considered invariance is the group of image content-preserving transformations (e.g. image flipping, 90-degree rotations, etc). Under the assumption that this group is restricted to consist of orthogonal matrices, the paper proves that a linear classifier trained on the resulting invariant features itself becomes invariant. The paper further shows how to perform PCA and k-means on these invariant features.

**Audience:**

Yes

**Broader Impact Concerns:**

I have no such concerns.

**Claims And Evidence:**

Yes

**Requested Changes:**

In line 3 of the introduction it states that $F = \frac{1}{N}\sum_{n=1}^N F(x_n) \in \mathbb{R}^{d_{global}}$. From this I deduce that $F$ is both a function $F: \mathbb{R}^{d_{local}} \rightarrow \mathbb{R}^{d_{global}}$ *and* a vector $F \in \mathbb{R}^{d_{global}}$. Already, at this stage, I am stuck, because those two seem to contradict each other.

The main result of the paper is Theorem 1. I found this to be highly underspecified to the degree that I do not know what the theorem shows. Specifically:
* The training data is defined as $\\{(v_m, y_m)\\}_{m=1}^M \in \mathbb{R}^d \times \mathcal{C}$. I assume that it's rather $(v_m, y_m) \in \mathbb{R}^d \times \mathcal{C}$, but even so: what is $\mathcal{C}$? This space does not appear to be defined.
* The loss function $l$, can that be any function? The text seems to indicate some notion of convexity of required, but I am unable to tell from the theorem precisely what is required (if anything).
* I did not understand the meaning of the sentence:
> ... which implies that $\\{(v_m, y_m)\\}_{m=1}^M \in \mathbb{R}^d \times \mathcal{C}$ exhibits the same distribution as $\\{(\pi(g)v_m, y_m)\\}_{m=1}^M \in \mathbb{R}^d \times \mathcal{C}$ for any $g$.
(apologies for formatting; I could not get Openreview to behave)
* In the proof, I did not understand the step from the second to the third line of Eq 3.
* In the last line of Eq 3, $w^{(1)}$ and $v_m^{(1)}$ does not appear to be defined.

All in all, I do not understand the statement in Theorem 1 nor do I understand its proof. This may be an issue on my side, but I see a significant lack of precision.

As a side-remark, I enjoyed the section "Group consisting of identity mapping and image flipping" in the appendix and I would recommend that it is moved to the main paper as it eased my second reading of the paper.

*Remark:*
Below I answered No to Claims and Evidence; this answer is based on my lack of understanding of the theory. I'm happy to change this in case a more precise paper is submitted.
*update:* After having read the revised version of the paper, I have changed this to Yes as the paper is now sufficiently clear.

*Remark:*
Below I answer Yes to Audience, but this is highly speculative as I am not the intended reader group, but I naively assume one such exist.

**Strengths And Weaknesses:**

## Strengths ##
As far as I can tell, the presented results appear novel (but I am not an expert in this particular niche), and empirical results seem to suggest modest improvements in hold-out predictive performance when enforcing invariant features. I also found the example in Fig. 2 to be quite illustrative: since the gradCam explanation algorithm is not equivariant to e.g. image flipping, then the associated explanations are not. The invariant features nicely improve this situation.

## Weaknesses ##
My main concern with the paper is that I find it very difficult to read. I am not part of the intended audience, so I understand that the paper is not aimed directly at me. Yet, I had to read the paper several times to perform this review, and I still struggle with the elementary aspects of the work. Consequently, the remainder of my review will focus on highlighting places where I got stuck (see Requested Changes below).

---

> ### Author Response · Authors · 2023-03-27
> **Author Response**
>
> We thank the reviewer for their thoughtful feedback.
>
> We will revise the manuscript following the comment.
>
> We will update the pdf after all the reviews are available, as the submission guideline (https://www.jmlr.org/tmlr/editorial-policies.html) suggests.
>
>
> Q1 About the expression $F=\frac{1}{N}\sum_{n=1}^N{F(x_n)}$.
>
> A1 Yes, $F$ indicates the global feature in the left side and indicates the function that maps local features to the global feature in the right side. This is confusing and we will change to $F=\frac{1}{N}\sum_{n=1}^N{\tilde{F}(x_n)}$.
>
> Q2 About Theorem 1.
>
> Q2a What is $\mathcal{C}$?
>
> A2a $\mathcal{C}$ denotes the label space. We will add the description.
>
> Q2b About the requirement for $l$.
>
> A2b $l$ is a loss function that calculates the loss value given the predicted label score and true label such as hinge loss. We require $l$ to be convex with respect to $\langle w, v_m\rangle_{\mathbb{R}}$ so that ${\frac{1}{M|G|}\sum_{m=1}^M\sum_{g\in G}{l\left(\mathrm{Re}\left(\sum_{t=1}^T \langle \tau_t(g) w^{(t)}, v_m^{(t)}\rangle_{\mathbb{C}}\right),y_m\right)}} \geq\frac{1}{M}\sum_{m=1}^M{l\left(\sum_{t=1}^T \mathrm{Re}\left(\left\langle\frac{1}{|G|}\sum_{g\in G}\tau_t(g) w^{(t)}, v_m^{(t)}\right\rangle_{\mathbb{C}}\right),y_m\right)}$ holds.
> This is the consequence of Jensen’s inequality with respect to the average over $g \in G$.
> We do not assume additional restriction on $l$.
>
> Q2c About the sentence "which implies that $\{(v_m,y_m)\}_{m=1}^M \in \mathbb{R}^d \times  \mathcal{C}$ exhibits the same distribution as $\{(\pi(g)v_m,y_m)\}\_{m=1}^M \in \mathbb{R}^d \times \mathcal{C}$ for any $g$".
>
> A2c We intended that $\{(v_m,y_m)\}\_{m=1}^M$ and $\{(\pi(g)v_m,y_m)\}\_{m=1}^M$ become the same multiset and therefore $\sum_{m=1}^M f(v_m,y_m) = \sum_{m=1}^M f(\pi(g) v_m,y_m)$ holds for any $f, g$.
> This assumption corresponds to the situation where we apply data augmentation to construct $\{(v_m,y_m)\}\_{m=1}^M$. For example, given the original dataset $\{(u_o,y_o)\}\_{o=1}^O$ and we construct the augmented dataset $\{(v_m,y_m)\}\_{m=1}^M$ as $\{(\pi(g) u_o,y_o)\}\_{o=1, g\in G}^O$, this augmented dataset is invariant under the action of $\pi(g)$ for any $g$.
>
> Q2d About the derivation of ${\frac{1}{M}\sum_{m=1}^M{l\left(\mathrm{Re}\left(\sum_{t=1}^T \langle w^{(t)}, v_m^{(t)}\rangle_{\mathbb{C}}\right),y_m\right)}}={\frac{1}{M|G|}\sum_{m=1}^M\sum_{g\in G}{l\left(\mathrm{Re}\left(\sum_{t=1}^T \langle w^{(t)}, \tau_t(g^{-1})v_m^{(t)}\rangle_{\mathbb{C}}\right),y_m\right)}}$.
>
> A2d It follows from the fact that $\sum_{m=1}^M f(v_m,y_m) = \sum_{m=1}^M f(\pi(g) v_m,y_m)$ holds for any $f, g$ as described in A2c. We apply this equation to all $g \in G$ and then average over $G$ to obtain the result.
>
> Q2e About $w^{(\textbf{1})}, v_m^{(\textbf{1})}$.
>
> A2e $w^{(\textbf{1})}$ and $v_m^{(\textbf{1})}$ correspond to the vector that we project $w$ and $v_m$ onto the subspace correspond to the trivial representation with respect to the irreducible decomposition. The last equality follows from the fact that $\tau_t(g)=1$ for trivial representation and $\sum_{g\in G}\tau_t(g) = 0$ for the other representations. We will add the explanation.
>
> Q3 About the section "Group consisting of identity mapping and image flipping" in the appendix.
>
> A3 We will move the section between Section 3 and 4.

---

> > ### Comment · Reviewer_xXoe · 2023-05-23
> > **Thanks**
> >
> > Thanks for the clarifications. I went through the updated manuscript and found it significantly more approachable (for me) then the initial version. I much appreciate these changes.

---

### Review · Reviewer_y3WA · 2023-04-25

**Summary Of Contributions:**

The authors propose a method of augmenting feature maps from static or learned linear-nonlinear neural network layers to encode invariance to standard identity-preserving transformations. They provide theoretical as well as experimental support for their ideas, which together suggest that such a method can lead to improved performance over no augmentation at all. Additionally, they compare their method against augmenting training data with the same transformations, and provide results suggesting that their method achieves approximately equivalent performance with lower training time requirements.

**Audience:**

Yes

**Broader Impact Concerns:**

No concerns.

**Claims And Evidence:**

Yes

**Requested Changes:**

**Easier:** These suggestions should be pretty easy to include in the final revision.

* You might consider how your contributions fit in with prior work from Ramakrishna Kakarala [1], Rajesh Rao [2-3], Bruno Olshausen [4-6], and colleagues. To be clear, I think your background section is quite comprehensive, as this is a fairly large research area. But in my opinion these works have carved out unique perspectives on the problem.

* You compare against augmenting the training data with the same transformations and note that your method has an improved training time. Isn't there alternatively an additional cost of combining the transformed feature representations at each activity map in your method? How do they actually compare? I think some numbers comparing the compute time or flops between the two methods would be interesting. Also, could you add a row repeating the scores of your methods without augmented training data to Table 5? This will make it easier to compare.

* I'm curious if you have a suggestion for why the accuracy trends in Table 3 are not consistent between the Test Accuracy and Augmented Test Accuracy conditions. Specifically, FMD & UIUC go up; DTD & CUB go down; and Cars stay the same.

**More involved:** These next suggestions might be a lot more work (re-running experiments; re-training networks with a new loss; implementing an alternative published analysis method). I think the paper is publication-worthy without answering them fully. However, I will leave it to the chair to require any of the changes in the context of other reviews.

* I think the standard practice for reporting ML accuracies is to report mean & variance across random seeds (e.g. for weight initialization & data shuffles). It helps to understand how much your accuracy gains. For example, Table 3 Test Accuracy for iBP on FMD is 0.08% better. Is that significant?

* I would be interested to see some more information on how you augmented the training data to obtain the results in Table 5. How does your method compare to [7]? Specifically, I'm interested to know if adding a regularization loss like they describe in eq 5 would improve the data augmentation results. [7] seems to suggest that this is necessary in addition to augmenting the data.

* I don't really see how the GradCAM results support your claims. You note that "The shape of the heatmap is relatively preserved with the proposed invariant coding method." I don't see this. The variance in the heatmap shape seems about the same between iSQRT-COV (Resnet50) (left column) and the proposed method. I didn’t notice much of a difference in the appendix figure either. Is there a way that you can quantify this? Perhaps some sort of intersection over union. You may also consider using GradCam++ [8]; the improved gradient masks might make your point more clear.

**Minor suggestions / typo edits:**

* Abstract: "explicitly consider group action" --> "explicitly considers group action"

* Bottom of page 1: "Therefore, this assumption is valid." --> I didn't follow this argument. I would appreciate it if you put a little more time into this part. It is a good section because it lays out a high-level description of your assumptions.

* Bottom of page 2, first contribution: Can you reword this contribution? We know that orthogonal perturbations to a linear classifier's weights will result in no difference in activation levels, because of linearity. I don't think this is what you're saying, but it is not clear to me how what you are saying here is different.

* Top of page 6, "Section Section 4.3" --> "Section 4.3"

* Bottom of page 13, "Table Table 8" --> "Table 8"

**Citations:**

[1] R. Kakarala, “The bispectrum as a source of phase-sensitive invariants for fourier descriptors:
A group-theoretic approach,” Journal of Mathematical Imaging and Vision, vol. 44, no. 3,
pp. 341–353, 2012.

[2] R. P. Rao and D. L. Ruderman, “Learning lie groups for invariant visual perception,” Advances
in neural information processing systems, pp. 810–816, 1999.

[3] X. Miao and R. P. Rao, “Learning the lie groups of visual invariance,” Neural computation,
vol. 19, no. 10, pp. 2665–2693, 2007.

[4] B. J. Culpepper and B. A. Olshausen, “Learning transport operators for image manifolds.,” in
NIPS, 2009, pp. 423–431.

[5] J. Sohl-Dickstein, C. M. Wang, and B. A. Olshausen, “An unsupervised algorithm for learning
lie group transformations,” arXiv preprint arXiv:1001.1027, 2010.

[6] H. Y. Chau, F. Qiu, Y. Chen, and B. Olshausen, “Disentangling images with lie group transformations and sparse coding,” arXiv preprint arXiv:2012.12071, 2020.

[7] G. Benton, M. Finzi, P. Izmailov, and A. G. Wilson, “Learning invariances in neural networks
from training data,” Advances in neural information processing systems, vol. 33, pp. 17 605–
17 616, 2020.

[8] Chattopadhay, A., Sarkar, A., Howlader, P., & Balasubramanian, V. N. (2018, March). Grad-cam++: Generalized gradient-based visual explanations for deep convolutional networks. In 2018 IEEE winter conference on applications of computer vision (WACV) (pp. 839-847). IEEE.


**Strengths And Weaknesses:**

The ideas and results are presented well. The authors provided an extensive set of experimental results to support their position, including ablation studies. In general, I am convinced that the performance gains they see are real. They also provide theoretical results that support their claims in an idealistic setting.

I don't see any weaknesses that should prevent the paper from being published.

---

> ### Author Response · Authors · 2023-05-17
> **Author Response**
>
> We thank the reviewer for their thoughtful feedback.
>
> We will revise the manuscript following the comment.
>
> Q1 About the related work.
>
> A1 We will add the discussion. The bispectrum invariant proposed in [1] relates to invariant Bilinear Pooling using SO(3) group. [2 - 7] are the work to learn the transformation group given invariant, while our purpose is to calculate invariant given transformation group.
>
> Q2 About computation complexity.
>
> A2 The discussion in Theorem 1 can be applicable for learning the linear classifier after the global features are obtained. In general, the additional computational cost required by the proposed feature coding is much smaller than the local feature extraction part and classifier training part. For example, in the experiment with fixed local features (Section 5.1), when we use ‘Intel Xeon E5-2698v4 x2 20 Core, 2.2 GHz’ CPU, it takes 13 seconds to extract the training features and 61 seconds to learn SVM to train BP on UIUC (108 training data, 18 categories), 9.4 seconds to extract the training features and 2.1 seconds to learn SVM to train Inv BP on UIUC in Table 3 and 81 seconds to extract the training features and 130 seconds to learn SVM to train BP on the augmented UIUC in Table 6.
>
> Q3 About scores of the proposed method in Table 6.
>
> A3 We will add the scores corresponding to the proposed invariant methods..
>
> Q4 About the difference between test accuracy and augmented test accuracy.
>
> A4 Related to A5, the differences are at most 0.2 and we think within the randomness considering standard deviation.
>
> Q5 About the variance.
>
> A5 We conducted the experiment 10 times for FMD, DTD and UIUC in Section 5.1 and CUB, Cars and Aircraft in Section 5.2 following the previous work. We will add standard deviations.
> As for iBP on FMD in Table 3, iBP demonstrates 81.38 $\pm$ 1.38 while the proposed Inv iBP demonstrates 83.46 $\pm$ 1.27 where 1.38 and 1.27 are unbiased standard deviations.
> With two-sample t-test, there exists difference in accuracy with p-value 0.0025.
>
> Q6 About augmentation method.
>
> A6 We apply all the transformations and make the number of training samples $|G|$ times as large. Mathematically, given the original dataset $\{(u_o,y_o)\}\_{o=1}^O$ and we construct the augmented dataset $\{(v_m,y_m)\}\_{m=1}^M$ as $\{(\pi(g) u_o,y_o)\}_{o=1, g\in G}^O$. As for the experiment corresponding to Table 6, the local feature extractor is fixed and only the feature coding function and classifier are trained and optimization method is not based on SGD. Therefore, it is difficult to add the regularizer in [7]. However, there is a possibility that the features are made close indirectly through the training of PCA and K-means. Furthermore, in the ideal setting, the difference in features are learned to be ignored by the classifier even without additional regularizer as Theorem 1 explains.
>
> Q7 About GradCAM.
>
> A7 We will replace the figure with GradCAM++. As for the quantitative evaluation, we evaluate the distance between the l1-normalized heatmaps using maximum mean discrepancy with gaussian kernel function $\exp(-\frac{\|r_1-r_2\|^2}{\sigma^2})$, where $r_1$, $r_2$ are 2d positions of each pixel and we set $\sigma=0.1$. From the tables below, we can see that the proposed method shows lower discrepancy.
>
> \begin{array}{l|cc}
>   & orig\leftrightarrow flip &  orig\leftrightarrow rotation \\\\ \hline
>   iSQRT-COV (Resnet50)   & 0.021   & 0.031 \\\\
>   iSQRT-COV (equivariant Resnet50)   & 0.068   & 0.11 \\\\
>   Inv iSQRT-COV (equivariant Resnet50) (ours)  & 0.017   & 0.022 \\\\ \hline
> \end{array}
> corresponds to Figure 5.
>
> \begin{array}{l|cc}
>   & orig\leftrightarrow flip &  orig\leftrightarrow rotation \\\\ \hline
>   iSQRT-COV (Resnet50)   & 0.027   & 0.023 \\\\
>   iSQRT-COV (equivariant Resnet50)   & 0.050   & 0.048 \\\\
>   Inv iSQRT-COV (equivariant Resnet50) (ours)  & 0.017   & 0.014 \\\\ \hline
> \end{array}
> corresponds to Figure 6.
>
> \begin{array}{l|cc}
>   & orig\leftrightarrow flip &  orig\leftrightarrow rotation \\\\ \hline
>   iSQRT-COV (Resnet50)   & 0.0082   & 0.014 \\\\
>   iSQRT-COV (equivariant Resnet50)   & 0.012   & 0.013 \\\\
>   Inv iSQRT-COV (equivariant Resnet50) (ours)  & 0.0082   & 0.0092 \\\\ \hline
> \end{array}
> corresponds to Figure 7.
>
> Q8 About the bottom of page 1.
>
> A8 By "Therefore, this assumption is valid.", we intended that the orthogonal assumption is not so restrictive, since the considered transformations can often be represented as the permutation between the local features.
>
> Q9 About the bottom of page 2.
>
> A9 The first contribution corresponds to Theorem 1. We will rewrite to “We prove that when we train the linear classifier on the space where the group acts orthogonally, the learned weight lies in the invariant subspace.”

---

> > ### Comment · Reviewer_y3WA · 2023-05-23
> > **Thank you**
> >
> > Thank you for addressing my concerns and performing the extra work. After looking over the paper again I am happy with the current state. It is a great piece of work, well done.

---

### Review · Reviewer_ZzG6 · 2023-05-09

**Summary Of Contributions:**

The manuscript proposes variants of the feature coding method which calculates a single global feature by aggregating the local feature extracted from an image.  It proposes 1) group-invariant extensions of feature coding methods, including Invariant PCA and Invariant K-means, and provides 2) a proof that the optimal solution of L2-regularized convex loss minimization, i.e., a linear classifier, is group-invariant when trained on the space where the group acts orthogonally. From the experiments conducted on image recognition datasets, the proposed methods improved the recognition performance compared to the baselines which produce non-invariant features.

**Audience:**

Yes

**Claims And Evidence:**

Yes

**Requested Changes:**

It would be better to add formal definitions or concrete examples for some notations at the place where the notations are first introduced. For example, what would the definitions of “group representations”, “group action” and “representations” be? In addition, about “g and G” in Section 3, I think the authors could introduce the D4 groups here as examples of “g and G”.

Minor: “\pi(g)and\sigama(g)” and “\piand\sigama”: space missing? >> “\pi and \sigama”?

It would be minor, but is not the loss in eq. (2) the form of multiclass classification losses? In linear multiclass classification, we need |C| number of weight parameters (w_{1, … ,|C|} unless the authors want to solve ordered multiclass classification problems. The descriptions are not clear enough.
In Section 4.5 and Section 5.2, it is not clear what kind of the loss function is used here. In addition, the parameter w should be a matrix whose column size is the same as the number of classes |C|.

About Theorem 1.
1. What is the formal definition of “T” in the eq. (3)?
2. It is not clear how v_m is connected to the local feature x_n. v_m is the global feature F?
3. Please include the explicit (formal) definition of the statement “The solution of the L2-regulazed convex loss minimization is G-invariant”?
4. “Furthermore, w^{(t)}s and x_{i}^{(t)} are orthogonal”: x_{i}^{(t)} is a typo? It was not mentioned in Theorem 1 before.
5. It is not clear the main point of Theorem 1. It is difficult to me to connect Theorem 1 with the generalization error. Could you elaborate more about the connection between Theorem 1 and generalization error? I am also wondering how practical this connection would be. What I mean, for example, this connection really explains why the proposed method worked better than the baselines in the experimental results?

**Strengths And Weaknesses:**

Strengths
The experiments using image recognition datasets and simple examples (for example Table 3, 4 and Figure 2) show that the proposed methods (based on invariant feature coding) are able to capture invariant image content while the baseline are not.

Weaknesses
1. The current version of the manuscript did not provide any complexity analyses of the proposed method. It even did not state about the computational environment where the experiments were done. It is not clear how much computational resources the proposed method would take compared to existing methods.
2. The manuscript could be improved in terms of presentation. It also seems that some details about the experimental settings are missing. Please see the detailed comments in “Requested Changes”.
3. The datasets used in the experiments are not large. Even UICU data includes around 12 images per each category in average. I am not sure if this size of the data sets is large enough to validate the performance of certain proposed models in this specific research area.

---

> ### Author Response · Authors · 2023-05-17
> **Author Response**
>
> We thank the reviewer for their thoughtful feedback.
>
> We will revise the manuscript following the comment.
>
> Q1 About computation time.
>
> A1 As for the experiment with fixed local features setting in Section 5.1.1, when we use ‘Intel Xeon E5-2698v4 x2 20 Core, 2.2 GHz’ CPU, it takes 13 seconds to extract the training features and 61 seconds to learn SVM to train BP on UIUC (108 training data, 18 categories), 9.4 seconds to extract the training features and 2.1 seconds to learn SVM to train Inv BP on UIUC in Table 3 and 81 seconds to extract the training features and 130 seconds to learn SVM to train BP on the augmented UIUC in Table 5. As for the end-to-end setting in Section 5.2.1, when we use 8 A100 GPUs, it takes 0.17 seconds/batch to train iSQRT-COV (Resnet50), 0.72 seconds/batch to train iSQRT-COV (equivariant Resnet50) and 0.45 seconds/batch to train the proposed Inv iSQRT-COV (equivariant Resnet50) on ImageNet. Though equivariant Resnet50 models takes more time than the original Resnet50, which mainly results from feature extractor module instead of the feature coding module we target at, we can reduce the computation time by the proposed invariant coding method compared to iSQRT-COV (equivariant Resnet50). The reduced complexity arises from (i) reduction of the feature dimension that we calculate the bilinear feature (ii) reduction of the size of the final fully-connected layer.
>
>
> Q2 About writing.
> Q2a About Section 3.
>
> A2a Following the comment of xXoe, we will move the section "Group consisting of identity mapping and image flipping" in the appendix to Section 3 to give the example.
>
> Q2b About the weight in Theorem 1.
>
> A2b We consider multi-class setting, but Eq. (2) becomes binary classification. We will fix it.
>
> Q2c About the loss in end-to-end setting.
>
> A2c We consider the convex loss in Section 4.5, but in Section 5.2, we used soft-max cross entropy loss to match the setting with the previous work. To be precise, the setting in Section 5.2 does not satisfy the assumption in Theorem 1.
>
> Q2d About Theorem 1
>
> A2d $T$ indicates the index of the irreducible representations. $v_m$ intends the global feature in the paper. However, Theorem 1 can be applicable whenever the assumption holds. “The solution of the L2-regulazed convex loss minimization is G-invariant” mathematically means that the solution $w$ satisfies $\pi(g) w = w$ for any $g$. $x_i^{(t)}$ should be replaced with $v_m^{(t)}$. Since $w^{(t)}$ and $v_m^{(t)}$ with different $t$ lie in different subspace with respect to the irreducible decomposition, they are orthogonal. As for generalization error, though the optimal solution is the same, there are two possibilities that the invariant feature works better. The first is that it is easier to optimize the classifier, since the feature dimension and the number of samples required is small. Second is that since the feature dimension becomes smaller, we can use larger size of code word for clustering-based methods.
>
> Q3 About the dataset size.
>
> A3 Though UIUC is relatively small, the dataset used in Section 5 are standard image recognition benchmarks. Furthermore, ImageNet is one of the standard large-scale image recognition benchmarks.

---

### Author Response · Authors · 2023-05-17
**Revision**

We thank again all the reviewers for their constructive comments.

We updated the manuscript following the comments.
Below we summarize the main updates.
We would be glad if the modification is satisfactory.

- We moved the example section in the appendix to Section 3.2, Section 4.1.2 and “Illustrative example of invariant k-means and VLAD” paragraph in Section 4.3 (ZzG6 Q2a and xXoe Q3).

- We added the discussion to Theorem 1 (ZzG6 Q2b, Q2d, xXoe Q2).

- We added standard deviation to the experimental results (y3WA Q5).

- We used GradCAM++ with quantitative comparison using maximum mean discrepancy in Figure 5,6 and 7 (y3WA Q7).

- We added the discussion about implementation and computation time in Section 5.1.1 and 5.2.1 (ZzG6 Q1, y3WA Q2).

- We added the related work (y3WA Q1).

- We added the explanation (y3WA Q3, Q8, Q9, xXoe Q1).

---

### Decision · Action_Editors · 2023-06-07

**Recommendation:** Accept as is

**Comment:**

In general, the paper is well written and revised. One reviewer suggested to include computational complexities of considered methods using big O notation. Please take it into account for the final version.

**Audience:**

The method on invariant feature coding, proposed in this paper, may be of interests for wider community.

**Claims And Evidence:**

This paper presents a feature coding method that exploits invariance. The ideas and results are clearly presented. The authors did a good job in revising the paper, taking initial reviews into account. Experiments well support the main results. The paper can be accepted as it is now.